# A Scottish provenance for the Altar Stone of Stonehenge

Anthony J. I. Clarke[1 ✉], Christopher L. Kirkland[1], Richard E. Bevins[2], Nick J. G. Pearce[2], Stijn Glorie[3] & Rob A. Ixer[4]

Understanding the provenance of megaliths used in the Neolithic stone circle at Stonehenge, southern England, gives insight into the culture and connectivity of prehistoric Britain. The source of the Altar Stone, the central recumbent sandstone megalith, has remained unknown, with recent work discounting an Anglo-Welsh Basin origin[1,2]. Here we present the age and chemistry of detrital zircon, apatite and rutile grains from within fragments of the Altar Stone. The detrital zircon load largely comprises Mesoproterozoic and Archaean sources, whereas rutile and apatite are dominated by a mid-Ordovician source. The ages of these grains indicate derivation from an ultimate Laurentian crystalline source region that was overprinted by Grampian (around 460 million years ago) magmatism. Detrital age comparisons to sedimentary packages throughout Britain and Ireland reveal a remarkable similarity to the Old Red Sandstone of the Orcadian Basin in northeast Scotland. Such a provenance implies that the Altar Stone, a 6 tonne shaped block, was sourced at least 750 km from its current location. The difficulty of long-distance overland transport of such massive cargo from Scotland, navigating topographic barriers, suggests that it was transported by sea. Such routing demonstrates a high level of societal organization with intra-Britain transport during the Neolithic period.

Stonehenge, the Neolithic standing stone circle located on the Salisbury Plain in Wiltshire, England, offers valuable insight into prehistoric Britain. Construction at Stonehenge began as early as 3000 BC, with subsequent modifications during the following two millennia[3,4]. The megaliths of Stonehenge are divided into two major categories: sarsen stones and bluestones (Fig. 1a). The larger sarsens comprise duricrust silcrete predominantly sourced from the West Woods, Marlborough, approximately 25 km north of Stonehenge[5,6]. Bluestone, the generic term for rocks considered exotic to the local area, includes volcanic tuff, rhyolite, dolerite and sandstone lithologies[4] (Fig. 1a). Some lithologies are linked with Neolithic quarrying sites in the Mynydd Preseli area of west Wales[7,8]. An unnamed Lower Palaeozoic sandstone, associated with the west Wales area on the basis of acritarch fossils[9], is present only as widely disseminated debitage at Stonehenge and possibly as buried stumps (Stones 40g and 42c).

The central megalith of Stonehenge, the Altar Stone (Stone 80), is the largest of the bluestones, measuring 4.9 × 1.0 × 0.5 m, and is a recumbent stone (Fig. 1b), weighing 6 t and composed of pale green micaceous sandstone with distinctive mineralogy[1,2,10] (containing baryte, calcite and clay minerals, with a notable absence of K-feldspar) (Fig. 2).

Previous petrographic work on the Altar Stone has implied an association to the Old Red Sandstone[10–12] (ORS). The ORS is a late Silurian to Devonian sedimentary rock assemblage that crops out widely throughout Great Britain and Ireland (Extended Data Fig. 1). ORS lithologies are dominated by terrestrial siliciclastic sedimentary rocks deposited in continental fluvial, lacustrine and aeolian environments[13]. Each ORS basin reflects local subsidence and sediment infill and thus contains proximal crystalline signatures[13,14].

Constraining the provenance of the Altar Stone could give insights into the connectivity of Neolithic people who left no written record[15]. When the Altar Stone arrived at Stonehenge is uncertain; however, it may have been placed within the central trilithon horseshoe during the second construction phase around 2620–2480 BC[3]. Whether the Altar Stone once stood upright as an approximately 4 m high megalith is unclear[15]; nevertheless, the current arrangement has Stones 55b and 156 from the collapsed Great Trilithon resting atop the prone and broken Altar Stone (Fig. 1b).

An early proposed source for the Altar Stone from Mill Bay, Pembrokeshire (Cosheston Subgroup of the Anglo-Welsh ORS Basin), close to the Mynydd Preseli source of the doleritic and rhyolitic bluestones, strongly influenced the notion of a sea transport route via the Bristol Channel[12]. However, inconsistencies in petrography and detrital zircon ages between the Altar Stone and the Cosheston Subgroup have ruled this source out[1,11]. Nonetheless, a source from elsewhere in the ORS of the Anglo-Welsh Basin was still considered likely, with an inferred collection and overland transport of the Altar Stone en route to Stonehenge from the Mynydd Preseli[1]. However, a source from the Senni Formation (Cosheston Subgroup) is inconsistent with geochemical and petrographic data, which shows that the Anglo-Welsh Basin is highly

[1]Timescales of Mineral Systems Group, School of Earth and Planetary Sciences, Curtin University, Perth, Western Australia, Australia. [2]Department of Geography and Earth Sciences, Aberystwyth University, Aberystwyth, UK. [3]Department of Earth Sciences, The University of Adelaide, Adelaide, South Australia, Australia. [4]Institute of Archaeology, University College London, London, UK. ✉e-mail: 20392091@student.curtin.edu.au

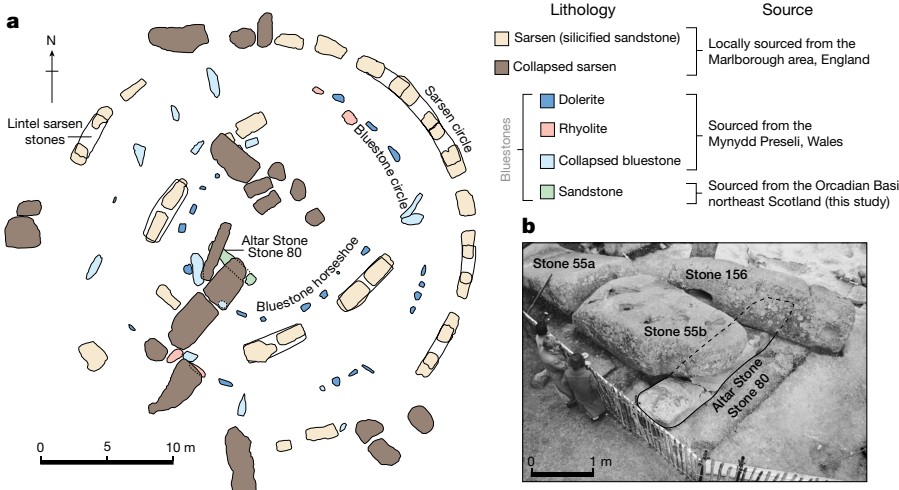

**Fig. 1 | The layout of Stonehenge and the appearance of the Altar Stone.**
**a**, Plan view of Stonehenge showing exposed constituent megaliths and their provenance. The plan of Stonehenge was adapted from ref. 6 under a CC BY 4.0 license. Changes in scale and colour were made, and annotations were added.

**b**, An annotated photograph shows the Altar Stone during a 1958 excavation. The Altar Stone photograph is from the Historic England archive. Reuse is not permitted.

unlikely to be the source[2]. Thus, the ultimate provenance of the Altar Stone had remained an open question.

Studies of detrital mineral grains are widely deployed to address questions throughout the Earth sciences and have utility in archaeological investigations[16,17]. Sedimentary rocks commonly contain a detrital component derived from a crystalline igneous basement, which may reflect a simple or complex history of erosion, transport and deposition cycles. This detrital cargo can fingerprint a sedimentary rock and its hinterland. More detailed insights become evident when a multi-mineral strategy is implemented, which benefits from the varying degrees of robustness to sedimentary transportation in the different minerals[18–20].

Here, we present in situ U–Pb, Lu–Hf and trace element isotopic data for zircon, apatite and rutile from two fragments of the Altar Stone collected at Stonehenge: MS3 and 2010K.240[21,22]. In addition, we present comparative apatite U–Pb dates for the Orcadian Basin from Caithness and Orkney. We utilize statistical tools (Fig. 3) to compare the obtained detrital mineral ages and chemistry (Supplementary Information 1–3) to crystalline terranes and ORS successions across Great Britain, Ireland and Europe (Fig. 4 and Extended Data Fig. 1).

## Laurentian basement signatures

The crystalline basement terranes of Great Britain and Ireland, from north to south, are Laurentia, Ganderia, Megumia and East Avalonia (Fig. 4a and Extended Data Fig. 1). Cadomia-Armorica is south of the Rheic Suture and encompasses basement rocks in western Europe, including northern France and Spain. East Avalonia, Megumia and Ganderia are partly separated by the Menai Strait Fault System (Fig. 4a). Each terrane has discrete age components, which have imparted palaeogeographic information into overlying sedimentary basins[13,14,23]. Laurentia was a palaeocontinent that collided with Baltica and Avalonia (a peri-Gondwanan microcontinent) during the early Palaeozoic Caledonian Orogeny to form Laurussia[14,24]. West Avalonia is a terrane that includes parts of eastern Canada and comprised the western margin of Avalonia (Extended Data Fig. 1).

Statistical comparisons, using a Kolmogorov–Smirnov test, between zircon ages from the Laurentian crystalline basement and the Altar Stone indicate that at a 95% confidence level, no distinction in provenance is evident between Altar Stone detrital zircon U–Pb ages and those from the Laurentian basement. That is, we cannot reject the null hypothesis that both samples are from the same underlying age distribution (Kolmogorov–Smirnov test: $P > 0.05$) (Fig. 3a).

Detrital zircon age components, defined by concordant analyses from at least 4 grains in the Altar Stone, include maxima at 1,047, 1,091, 1,577, 1,663 and 1,790 Ma (Extended Data Fig. 2), corresponding to known tectonomagmatic events and sources within Laurentia and Baltica, including the Grenville (1,095–980 Ma), Labrador (1,690–1,590 Ma), Gothian (1,660–1,520 Ma) and Svecokarellian (1,920–1,770 Ma) orogenies[25].

Laurentian terranes are crystalline lithologies north of the Iapetus Suture Zone (which marks the collision zone between Laurentia and Avalonia) and include the Southern Uplands, Midland Valley, Grampian, Northern Highlands and Hebridean Terranes (Fig. 4a). Together, these terranes preserve a Proterozoic to Archaean record of zircon production[24], distinct from the southern Gondwanan-derived terranes of Britain[20,26] (Fig. 4a and Extended Data Fig. 3).

Age data from Altar Stone rutile grains also point towards an ultimate Laurentian source with several discrete age components (Extended Data Fig. 4 and Supplementary Information 1). Group 2 rutile U–Pb analyses from the Altar Stone include Proterozoic ages from 1,724 to 591 Ma, with 3 grains constituting an age peak at 1,607 Ma, overlapping with Laurentian magmatism, including the Labrador and Pinwarian (1,690–1,380 Ma) orogenies[24]. Southern terranes in Britain are not characterized by a large Laurentian (Mesoproterozoic) crystalline age component[25] (Fig. 4b and Extended Data Fig. 3). Instead, terranes south of the Iapetus Suture are defined by Neoproterozoic to early Palaeozoic components, with a minor component from around two billion years ago (Figs. 3b and 4b).

U–Pb analyses of apatite from the Altar Stone define two distinct age groupings. Group 2 apatite U–Pb analyses define a lower intercept age of $1,018 \pm 24$ Ma ($n = 9$) (Extended Data Fig. 5), which overlaps, within uncertainty, to a zircon age component at 1,047 Ma, consistent with a Grenville source[25]. Apatite Lu–Hf dates at 1,496 and 1,151 Ma also imply distinct Laurentian sources[25] (Fig. 4b, Extended Data Fig. 6 and Supplementary Information 2). Ultimately, the presence of Grenvillian apatite in the Altar Stone suggests direct derivation from the Laurentian basement, given the lability of apatite during prolonged chemical weathering[20,27].

## Grampian Terrane detrital grains

Apatite and rutile U–Pb analyses from the Altar Stone are dominated by regressions from common Pb that yield lower intercepts of $462 \pm 4$ Ma

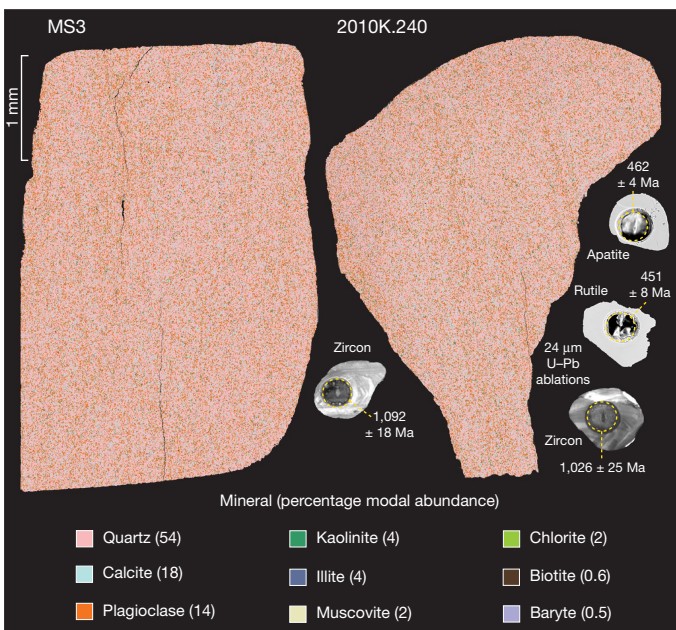

**Fig. 2 | False colour automated mineralogy maps from thin sections of the Altar Stone.** Minerals with a modal abundance above 0.5% are shown with compositional values averaged across both thin sections. U–Pb ablation pits from laser ablation inductively coupled plasma mass spectrometry (LA-ICP–MS) are shown with age (in millions of years ago, Ma), with uncertainty at the 2σ level.

(n = 108) and 451 ± 8 Ma (n = 83), respectively (Extended Data Figs. 4 and 5). A single concordant zircon analysis also yields an early Palaeozoic age of 498 ± 17 Ma. Hence, with uncertainty from both lower intercepts, Group 1 apatite and rutile analyses demonstrate a mid-Ordovician (443–466 Ma) age component in the Altar Stone. These mid-Ordovician ages are confirmed by in situ apatite Lu–Hf analyses, which define a lower intercept of 470 ± 29 Ma (n = 16) (Extended Data Fig. 6 and Supplementary Information 2).

Throughout the Altar Stone are sub-planar 100–200-µm bands of concentrated heavy resistive minerals. These resistive minerals are interpreted to be magmatic in origin, given internal textures (oscillatory zonation), lack of mineral overgrowths (in all dated minerals) (Fig. 2) and the igneous apatite trace element signatures[27] (Extended Data Fig. 7 and Supplementary Information 3). Moreover, there is a general absence of detrital metamorphic zircon grains, further supporting a magmatic origin for these grains.

The most appropriate source region for such mid-Ordovician grains within Laurentian basement is the Grampian Terrane of northeast Scotland (Fig. 4a). Situated between the Great Glen Fault to the north and the Highland Boundary Fault to the south, the terrane comprises Neoproterozoic to Lower Palaeozoic metasediments termed the Dalradian Supergroup[28], which are intruded by a compositionally diverse suite of early Palaeozoic granitoids and gabbros (Fig. 4a). The 466–443 Ma age component from Group 1 apatite and rutile U–Pb analyses overlaps with the terminal stages of Grampian magmatism and subsequent granite pluton emplacement north of the Highland Boundary Fault[28] (Fig. 4a).

Geochemical classification plots for the Altar Stone apatite imply a compositionally diverse source, much like the lithological diversity within the Grampian Terrane[28], with 61% of apatite classified as coming from felsic sources, 35% from mafic sources and 4% from alkaline sources (Extended Data Fig. 7 and Supplementary Information 3). Specifically, igneous rocks within the Grampian Terrane are largely granitoids, thus accounting for the predominance of felsic-classified apatite grains[29]. We posit that the dominant supply of detritus from

466–443 Ma came from the numerous similarly aged granitoids formed on the Laurentian margin[28], which are present in both the Northern Highlands and the Grampian Terranes[28] (Fig. 4a). The alkaline to calc-alkaline suites in these terranes are volumetrically small, consistent with the scarcity of alkaline apatite grains within the Altar Stone (Extended Data Fig. 7). Indeed, the Glen Dessary syenite at 447 ± 3 Ma is the only age-appropriate felsic-alkaline pluton in the Northern Highlands Terrane[30].

The Stacey and Kramers[31] model of terrestrial Pb isotopic evolution predicts a $^{207}Pb/^{206}Pb$ isotopic ratio ($^{207}Pb/^{206}Pb_i$) of 0.8601 for 465 Ma continental crust. Mid-Ordovician regressions through Group 1 apatite and rutile U–Pb analyses yield upper intercepts for $^{207}Pb/^{206}Pb_i$ of 0.8603 ± 0.0033 and 0.8564 ± 0.0014, respectively (Extended Data Figs. 4 and 5 and Supplementary Information 1). The similarity between apatite and rutile $^{207}Pb/^{206}Pb_i$ implies they were sourced from the same Mid-Ordovician magmatic fluids. Ultimately, the calculated $^{207}Pb/^{206}Pb_i$ value is consistent with the older (Laurentian) crust north of the Iapetus Suture in Britain[32] (Fig. 4a).

## Orcadian Basin ORS

The detrital zircon age spectra confirm petrographic associations between the Altar Stone and the ORS. Furthermore, the Altar Stone cannot be a New Red Sandstone (NRS) lithology of Permo-Triassic age. The NRS, deposited from around 280–240 Ma, unconformably overlies the ORS[14]. NRS, such as that within the Wessex Basin (Extended Data Fig. 1), has characteristic detrital zircon age components, including Carboniferous to Permian zircon grains, which are not present in the Altar Stone[1,23,26,33,34] (Extended Data Fig. 3).

An ORS classification for the Altar Stone provides the basis for further interpretation of provenance (Extended Data Figs. 1 and 8), given that the ORS crops out in distinct areas of Great Britain and Ireland, including the Anglo-Welsh border and south Wales, the Midland Valley and northeast Scotland, reflecting former Palaeozoic depocentres[14] (Fig. 4a).

Previously reported detrital zircon ages and petrography show that ORS outcrops of the Anglo-Welsh Basin in the Cosheston Subgroup[1] and Senni Formation[2] are unlikely to be the sources of the Altar Stone (Fig. 4a). ORS within the Anglo-Welsh Basin is characterized by mid-Palaeozoic zircon age maxima and minor Proterozoic components (Fig. 4a). Ultimately, the detrital zircon age spectra of the Altar Stone are statistically distinct from the Anglo-Welsh Basin (Fig. 3a). In addition, the ORS outcrops of southwest England (that is, south of the Variscan front), including north Devon and Cornwall (Cornubian Basin) (Fig. 4a), show characteristic facies, including marine sedimentary structures and fossils along with a metamorphic fabric[13,26], inconsistent with the unmetamorphosed, terrestrial facies of the Altar Stone[1,11].

Another ORS succession with published age data for comparison is the Dingle Peninsula Basin, southwest Ireland. However, the presence of late Silurian (430–420 Ma) and Devonian (400–350 Ma) apatite, zircon and muscovite from the Dingle Peninsula ORS discount a source for the Altar Stone from southern Ireland[20]. The conspicuous absence of apatite grains of less than 450 Ma in age in the Altar Stone precludes the input of Late Caledonian magmatic grains to the source sediment of the Altar Stone and demonstrates that the ORS of the Altar Stone was deposited prior to or distally from areas of Late Caledonian magmatism, unlike the ORS of the Dingle Peninsula[20]. Notably, no distinction in provenance between the Anglo-Welsh Basin and the Dingle Peninsula ORS is evident (Kolmogorov–Smirnov test: P > 0.05), suggesting that ORS basins south of the Iapetus Suture are relatively more homogenous in terms of their detrital zircon age components (Fig. 4a).

In Scotland, ORS predominantly crops out in the Midland Valley and Orcadian Basins (Fig. 4a). The Midland Valley Basin is bound between the Highland Boundary Fault and the Iapetus Suture and is located within the Midland Valley and Southern Uplands Terranes. Throughout

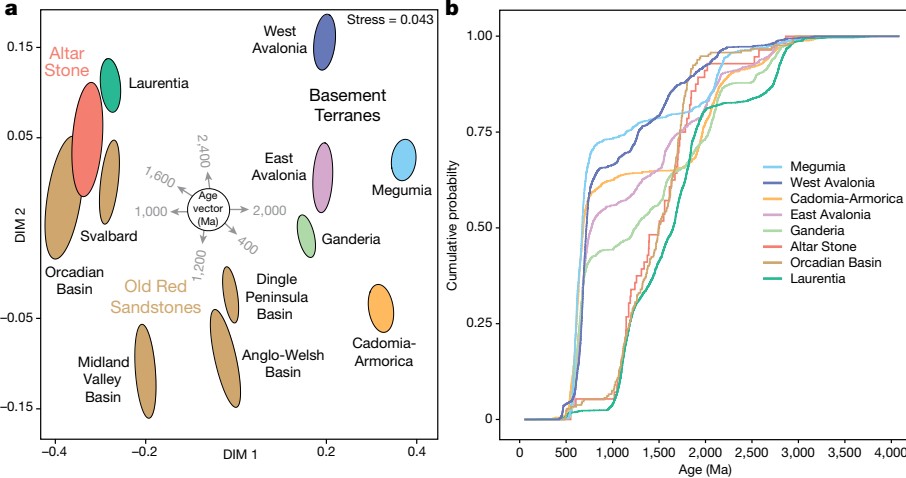

**Fig. 3 | Comparisons of detrital zircon U–Pb ages from the Altar Stone and crystalline terranes of Britain, Ireland, and Europe. a**, Multidimensional scaling (MDS) plot of concordant zircon U–Pb ages from the Altar Stone and comparative age datasets, with ellipses at the 95% confidence level[58]. DIM 1 and

DIM 2, dimensions 1 and 2. **b**, Cumulative probability plot of zircon U–Pb ages from crystalline terranes, the Orcadian Basin and the Altar Stone. For a cumulative probability plot of all ORS basins, see Extended Data Fig. 8.

Midland Valley ORS stratigraphy, detrital zircon age spectra broadly show a bimodal age distribution between Lower Palaeozoic and Mesoproterozoic components[35,36] (Extended Data Fig. 3). Indeed, throughout 9 km of ORS stratigraphy in the Midland Valley Basin and across the Sothern Uplands Fault, no major changes in provenance are recognized[36] (Fig. 4a). Devonian zircon, including grains as young as $402 \pm 5$ Ma from the northern ORS in the Midland Valley Basin[36], further differentiates this basin from the Altar Stone (Fig. 3a and Extended Data Fig. 3). The scarcity of Archaean to late Palaeoproterozoic zircon grains within the Midland Valley ORS shows that the Laurentian basement was not a dominant detrital source for those rocks[35]. Instead, ORS of the Midland Valley is primarily defined by zircon from 475 Ma interpreted to represent the detrital remnants of Ordovician volcanism within the Midland Valley Terrane, with only minor and periodic input from Caledonian plutonism[35].

The Orcadian Basin of northeast Scotland, within the Grampian and Northern Highlands terranes, contains a thick package of mostly Mid-Devonian ORS, around 4 km thick in Caithness and up to around 8 km thick in Shetland[14] (Fig. 4a). The detrital zircon age spectra from Orcadian Basin ORS provides the closest match to the Altar Stone detrital ages[25] (Fig. 3 and Extended Data Fig. 8). A Kolmogorov–Smirnov test on age spectra from the Altar Stone and the Orcadian Basin fails to reject the null hypothesis that they are derived from the same underlying distribution (Kolmogorov–Smirnov test: $P > 0.05$) (Fig. 3a). To the north, ORS on the Svalbard archipelago formed on Laurentian and Baltican basement rocks[37]. Similar Kolmogorov–Smirnov test results, where each detrital zircon dataset is statistically indistinguishable, are obtained for ORS from Svalbard, the Orcadian Basin and the Altar Stone.

Apatite U–Pb age components from Orcadian Basin samples from Spittal, Caithness (AQ1) and Cruaday, Orkney (CQ1) (Fig. 4a) match those from the Altar Stone. Group 2 apatite from the Altar Stone at $1,018 \pm 24$ Ma is coeval with a Grenvillian age from Spittal at $1,013 \pm 35$ Ma. Early Palaeozoic apatite components at $473 \pm 25$ Ma and $466 \pm 6$ Ma, from Caithness and Orkney, respectively (Extended Data Fig. 5 and Supplementary Information 1), are also identical, within uncertainty, to Altar Stone Group 1 ($462 \pm 4$ Ma) apatite U–Pb analyses and a Lu–Hf component at $470 \pm 28$ Ma supporting a provenance from the Orcadian Basin for the Altar Stone (Extended Data Fig. 6 and Supplementary Information 2).

During the Palaeozoic, the Orcadian Basin was situated between Laurentia and Baltica on the Laurussian palaeocontinent[14]. Correlations between detrital zircon age components imply that both Laurentia and Baltica supplied sediment into the Orcadian Basin[25,36]. Detrital

grains from more than 900 Ma within the Altar Stone are consistent with sediment recycling from intermediary Neoproterozoic supracrustal successions (for example, Dalradian Supergroup) within the Grampian Terrane but also from the Särv and Sparagmite successions of Baltica[25,36]. At around 470 Ma, the Grampian Terrane began to denude[28]. Subsequently, first-cycle detritus, such as that represented by Group 1 apatite and rutile, was shed towards the Orcadian Basin from the southeast[25].

Thus, the resistive mineral cargo in the Altar Stone represents a complex mix of first and multi-cycle grains from multiple sources. Regardless of total input from Baltica versus Laurentia into the Orcadian Basin, crystalline terranes north of the Iapetus Suture (Fig. 4a) have distinct age components that match the Altar Stone in contrast to Gondwanan-derived terranes to the south.

## The Altar Stone and Neolithic Britain

Isotopic data for detrital zircon and rutile (U–Pb) and apatite (U–Pb, Lu–Hf and trace elements) indicate that the Altar Stone of Stonehenge has a provenance from the ORS in the Orcadian Basin of northeast Scotland (Fig. 4a). Given this detrital mineral provenance, the Altar Stone cannot have been sourced from southern Britain (that is, south of the Iapetus Suture) (Fig. 4a), including the Anglo-Welsh Basin[1,2].

Some postulate a glacial transport mechanism for the Mynydd Preseli (Fig. 4a) bluestones to Salisbury Plain[38,39]. However, such transport for the Altar Stone is difficult to reconcile with ice-sheet reconstructions that show a northwards movement of glaciers (and erratics) from the Grampian Mountains towards the Orcadian Basin during the Last Glacial Maximum and, indeed, previous Pleistocene glaciations[40,41]. Moreover, there is little evidence of extensive glacial deposition in central southern Britain[40], nor are Scottish glacial erratics found at Stonehenge[42]. Sr and Pb isotopic signatures from animal and human remains from henges on Salisbury Plain demonstrate the mobility of Neolithic people within Britain[32,43–45]. Furthermore, shared architectural elements and rock art motifs between Neolithic monuments in Orkney, northern Britain, and Ireland point towards the long-distance movement of people and construction materials[46,47].

Thus, we posit that the Altar Stone was anthropogenically transported to Stonehenge from northeast Scotland, consistent with evidence of Neolithic inhabitation in this region[48,49]. Whereas the igneous bluestones were brought around 225 km from the Mynydd Preseli to Stonehenge[50] (Fig. 4a), a Scottish provenance for the Altar Stone

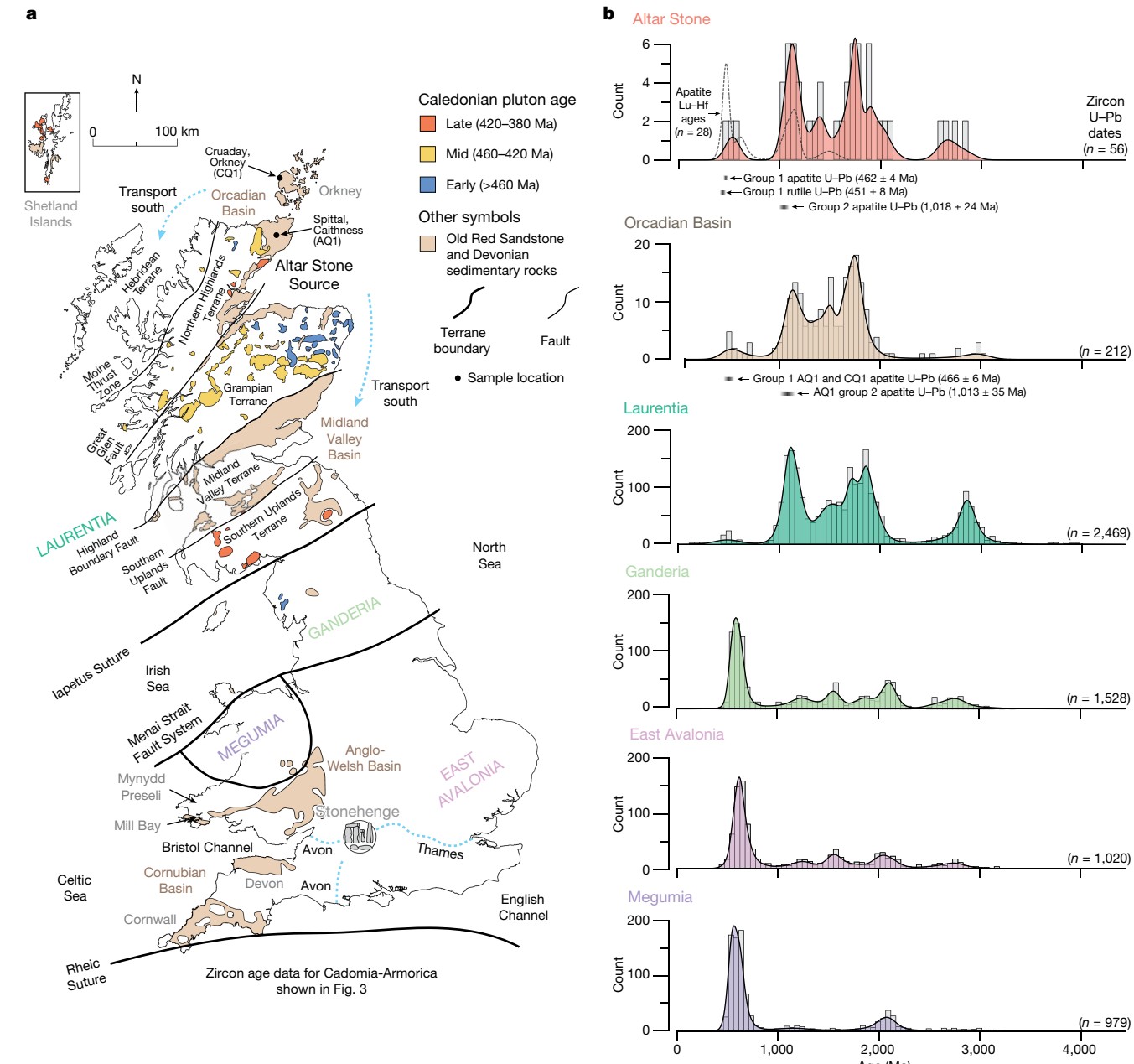

**Fig. 4 | The zircon age spectra of geological terranes and ORS basins of Britain compared with the Altar Stone. a**, Schematic map of Britain, showing outcrops of ORS and other Devonian sedimentary rocks, basement terranes and major faults. Potential Caledonian source plutons are colour-coded on the basis of age[28]. **b**, Kernel density estimate diagrams displaying zircon U–Pb age (histogram) and apatite Lu–Hf age (dashed line) spectra from the Altar Stone, the Orcadian Basin[25] and plausible crystalline source terranes. The apatite age components for the Altar Stone and Orcadian Basins are shown below their respective kernel density estimates. Extended Data Fig. 3 contains kernel density estimates of other ORS and New Red Sandstone (NRS) age datasets.

demands a transport distance of at least 750 km (Fig. 4a). Nonetheless, even with assistance from beasts of burden[51], rivers and topographical barriers, including the Grampians, Southern Uplands and the Pennines, along with the heavily forested landscape of prehistoric Britain[52], would have posed formidable obstacles for overland megalith transportation.

At around 5000 BC, Neolithic people introduced the common vole (*Microtus arvalis*) from continental Europe to Orkney, consistent with the long-distance marine transport of cattle and goods[53]. A Neolithic marine trade network of quarried stone tools is found throughout Britain, Ireland and continental Europe[54]. For example, a saddle quern, a large stone grinding tool, was discovered in Dorset and determined to have a provenance in central Normandy[55], implying the shipping of stone cargo over open water during the Neolithic. Furthermore, the river transport of shaped sandstone blocks in Britain is known from at

least around 1500 BC (Hanson Log Boat)[56]. In Britain and Ireland, sea levels approached present-day heights from around 4000 BC[57], and although coastlines have shifted, the geography of Britain and Ireland would have permitted sea routes southward from the Orcadian Basin towards southern England (Fig. 4a). A Scottish provenance for the Altar Stone implies Neolithic transport spanning the length of Great Britain.

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

## Methods

### Overview

This work analysed two 30-μm polished thin sections of the Altar Stone (MS3 and 2010K.240) and two sections of ORS from northeast Scotland (Supplementary Information 4). CQ1 is from Cruaday, Orkney (59° 04′ 34.2″ N, 3° 18′ 54.6″ W), and AQ1 is from near Spittal, Caithness (58° 28′ 13.8″ N, 3° 27′ 33.6″ W). Conventional optical microscopy (transmitted and reflected light) and automated mineralogy via a TESCAN Integrated Mineral Analyser gave insights into texture and mineralogy and guided spot placement during LA-ICP–MS analysis. A CLARA field emission scanning electron microscope was used for textural characterization of individual minerals (zircon, apatite and rutile) through high-resolution micrometre-scale imaging under both back-scatter electron and cathodoluminescence. The Altar Stone is a fine-grained and well-sorted sandstone with a mean grain size diameter of ≤300 μm. Quartz grains are sub-rounded and monocrystalline. Feldspars are variably altered to fine-grained white mica. MS3 and 2010K.240 have a weakly developed planar fabric and non-planar heavy mineral laminae approximately 100–200 μm thick. Resistive heavy mineral bands are dominated by zircon, rutile, and apatite, with grains typically 10–40 μm wide. The rock is mainly cemented by carbonate, with localized areas of barite and quartz cement. A detailed account of Altar Stone petrography is provided in refs. 1,59.

### Zircon isotopic analysis

**Zircon U–Pb methods.** Two zircon U–Pb analysis sessions were completed at the GeoHistory facility in the John De Laeter Centre (JdLC), Curtin University, Australia. Ablations within zircon grains were created using an excimer laser RESOlution LE193 nm ArF with a Laurin Technic S155 cell. Isotopic data was collected with an Agilent 8900 triple quadrupole mass spectrometer, with high-purity Ar as the plasma carrier gas (flow rate 1.l min[-1]). An on-sample energy of -2.3–2.7 J cm[-2] with a 5–7 Hz repetition rate was used to ablate minerals for 30–40 s (with 25–60 s of background capture). Two cleaning pulses preceded analyses, and ultra-high-purity He (0.68 ml min[-1]) and N₂ (2.8 ml min[-1]) were used to flush the sample cell. A block of reference mineral was analysed following 15–20 unknowns. The small, highly rounded target grains of the Altar Stone (usually <30 μm in width) necessitated using a spot size diameter of -24 μm for all ablations. Isotopic data was reduced using Iolite 4[60] with the U-Pb Geochronology data reduction scheme, followed by additional calculation and plotting via IsoplotR[61]. The primary matrix-matched reference zircon[62] used to correct instrumental drift and mass fractionation was GJ-1, 601.95 ± 0.40 Ma. Secondary reference zircon included Plešovice[63], 337.13 ± 0.37 Ma, 91500[64], 1,063.78 ± 0.65 Ma, OG1[65] 3,465.4 ± 0.6 Ma and Maniitsoq[66] 3,008.7 ± 0.6 Ma. Weighted mean U–Pb ages for secondary reference materials were within 2σ uncertainty of reported values (Supplementary Information 5).

**Zircon U–Pb results.** Across two LA-ICP–MS sessions, 83 U–Pb measurements were obtained on as many zircon grains; 41 were concordant (≤10% discordant), where discordance is defined using the concordia log distance (%) approach[67]. We report single-spot (grain) concordia ages, which have numerous benefits over conventional U–Pb/Pb–Pb ages, including providing an objective measure of discordance that is directly coupled to age and avoids the arbitrary switch between ²⁰⁶Pb/²³⁸U and ²⁰⁷Pb/²⁰⁶Pb. Furthermore, given the spread in ages (Early Palaeozoic to Archaean), concordia ages provide optimum use of both U–Pb/Pb–Pb ratios, offering greater precision over ²⁰⁶Pb/²³⁸U or ²⁰⁷Pb/²⁰⁶Pb ages alone.

Given that no direct sampling of the Altar Stone is permitted, we are limited in the amount of material available for destructive analysis, such as LA-ICP–MS. We collate our zircon age data with the U–Pb analyses[1] of FN593 (another fragment of the Altar Stone), filtered using

the same concordia log distance (%) discordance filter[67]. The total concordant analyses used in this work is thus 56 over 3 thin sections, each showing no discernible provenance differences. Zircon concordia ages span from 498 to 2,812 Ma. Age maxima (peak) were calculated after Gehrels[68], and peak ages defined by ≥4 grains include 1,047, 1,091, 1,577, 1,663 and 1,790 Ma.

For 56 concordant ages from 56 grains at >95% certainty, the largest unmissed fraction is calculated at 9% of the entire uniform detrital population[69]. In any case, the most prevalent and hence provenance important components will be sampled for any number of analyses[69]. We analysed all zircon grains within the spatial limit of the technique in the thin sections[70]. We used in situ thin-section analysis, which can mitigate against contamination and sampling biases in detrital studies[71]. Adding apatite (U-Pb and Lu-Hf) and rutile (U-Pb) analyses bolsters our confidence in provenance interpretations as these minerals will respond dissimilarly during transport.

**Comparative zircon datasets.** Zircon U–Pb compilations of the basement terranes of Britain and Ireland were sourced from refs. 20,26. ORS detrital zircon datasets used for comparison include isotopic data from the Dingle Peninsula Basin[20], Anglo-Welsh Basin[72], Midland Valley Basin[35], Svalbard ORS[37] and Orcadian Basin[25]. NRS zircon U–Pb ages were sourced from the Wessex Basin[33]. Comparative datasets were filtered for discordance as per our definition above[20,26]. Kernel density estimates for age populations were created within IsoplotR[61] using a kernel and histogram bandwidth of 50 Ma.

A two-sample Kolmogorov–Smirnov statistical test was implemented to compare the compiled zircon age datasets with the Altar Stone (Supplementary Information 6). This two-sided test compares the maximum probability difference between two cumulative density age functions, evaluating the null hypothesis that both age spectra are drawn from the same distribution based on a critical value dependent on the number of analyses and a chosen confidence level.

The number of zircon ages within the comparative datasets used varies from the Altar Stone (n = 56) to Laurentia (n = 2,469). Therefore, to address the degree of dependence on sample n, we also implemented a Monte Carlo resampling (1,000 times) procedure for the Kolmogorov–Smirnov test, including the uncertainty on each age determination to recalculate P values and standard deviations (Supplementary Information 7), based on the resampled distribution of each sample. The results from Kolmogorov–Smirnov tests, using Monte Carlo resampling (and multidimensional analysis), taking uncertainty due to sample n into account, also support the interpretation that at >95% certainty, no distinction in provenance can be made between the Altar Stone zircon age dataset (n = 56) and those from the Orcadian Basin (n = 212), Svalbard ORS (n = 619) and the Laurentian basement (Supplementary Information 7).

MDS plots for zircon datasets were created using the MATLAB script of ref. 58. Here, we adopted a bootstrap resampling (>1,000 times) with Procrustes rotation of Kolmogorov–Smirnov values, which outputs uncertainty ellipses at a 95% confidence level (Fig. 3a). In MDS plots, stress is a goodness of fit indicator between dissimilarities in the datasets and distances on the MDS plot. Stress values below 0.15 are desirable[58]. For the MDS plot in Fig. 3a, the value is 0.043, which indicates an "excellent" fit[58].

### Rutile isotopic analysis

**Rutile U–Pb methods.** One rutile U–Pb analysis session was completed at the GeoHistory facility in the JdLC, Curtin University, Australia. Rutile grains were ablated (24 μm) using a Resonetics RESOlution M-50A-LR sampling system, using a Compex 102 excimer laser, and measured using an Agilent 8900 triple quadrupole mass analyser. The analytical parameters included an on-sample energy of 2.7 J cm[-2], a repetition rate of 7 Hz for a total analysis time of 45 s, and 60 s of background data

capture. The sample chamber was purged with ultrahigh purity He at a flow rate of 0.68 l min$^{-1}$ and N$_2$ at 2.8 ml min$^{-1}$.

U–Pb data for rutile analyses was reduced against the R-10 rutile primary reference material[73] (1,091 ± 4 Ma). The secondary reference material used to monitor the accuracy of U–Pb ratios was R-19 rutile. The mean weighted $^{238}$U/$^{206}$Pb age obtained for R-19 was 491 ± 10 (mean squared weighted deviation (MSWD) = 0.87, $p(\chi^2)$ = 0.57) within uncertainty of the accepted age[74] of 489.5 ± 0.9 Ma.

Rutile grains with negligible Th concentrations can be corrected for common Pb using a $^{208}$Pb correction[74]. Previously used thresholds for Th content have included[75,76] Th/U < 0.1 or a Th concentration >5% U. However, Th/U ratios for rutile from MS3 are typically >1; thus, a $^{208}$Pb correction is not applicable. Instead, we use a $^{207}$-based common Pb correction[31] to account for the presence of common Pb. Rutile isotopic data was reduced within Iolite 4[60] using the U–Pb Geochronology reduction scheme and IsoplotR[61].

**Rutile U–Pb Results.** Ninety-two rutile U–Pb analyses were obtained in a U–Pb single session, which defined two coherent age groupings on a Tera–Wasserburg plot.

Group 1 constitutes 83 U–Pb rutile analyses, forming a well-defined mixing array on a Tera-Wasserburg plot between common and radiogenic Pb components. This array yields an upper intercept of $^{207}$Pb/$^{206}$Pb$_i$ = 0.8563 ± 0.0014. The lower intercept implies an age of 451 ± 8 Ma. The scatter about the line (MSWD = 2.7) is interpreted to reflect the variable passage of rutile of diverse grain sizes through the radiogenic Pb closure temperature at ~600 °C during and after magmatic crystallization[77].

Group 2 comprises 9 grains, with $^{207}$Pb corrected $^{238}$U/$^{206}$Pb ages ranging from 591–1,724 Ma. Three grains from Group 2 define an age peak[68] at 1,607 Ma. Given the spread in U–Pb ages, we interpret these Proterozoic grains to represent detrital rutile derived from various sources.

### Apatite isotopic analysis

**Apatite U–Pb methods.** Two apatite U–Pb LA-ICP–MS analysis sessions were conducted at the GeoHistory facility in the JdLC, Curtin University, Australia. For both sessions, ablations were created using a RESOlution 193 nm excimer laser ablation system connected to an Agilent 8900 ICP–MS with a RESOlution LE193 nm ArF and a Laurin Technic S155 cell ICP–MS. Other analytical details include a fluence of 2 J cm$^2$ and a 5 Hz repetition rate. For the Altar Stone section (MS3) and the Orcadian Basin samples (Supplementary Information 4), 24- and 20-µm spot sizes were used, respectively.

The matrix-matched primary reference material used for apatite U–Pb analyses was the Madagascar apatite (MAD-1)[78]. A range of secondary reference apatite was analysed, including FC-1[79] (Duluth Complex) with an age of 1,099.1 ± 0.6 Ma, Mount McClure[80,81] 526 ± 2.1 Ma, Otter Lake[82] 913 ± 7 Ma and Durango 31.44 ± 0.18[83] Ma. Anchored regressions (through reported $^{207}$Pb/$^{206}$Pb$_i$ values) for secondary reference material yielded lower intercept ages within 2$\sigma$ uncertainty of reported values (Supplementary Information 8).

**Altar Stone apatite U–Pb results.** This first session of apatite U–Pb of MS3 from the Altar Stone yielded 117 analyses. On a Tera–Wasserburg plot, these analyses form two discordant mixing arrays between common and radiogenic Pb components with distinct lower intercepts.

The array from Group 2 apatite, comprised of 9 analyses, yields a lower intercept equivalent to an age of 1,018 ± 24 Ma (MSWD = 1.4) with an upper intercept $^{207}$Pb/$^{206}$Pb$_i$ = 0.8910 ± 0.0251. The f$^{207}$% (the percentage of common Pb estimated using the $^{207}$Pb method) of apatite analyses in Group 2 ranges from 16.66–88.8%, with a mean of 55.76%.

Group 1 apatite is defined by 108 analyses yielding a lower intercept of 462 ± 4 Ma (MSWD = 2.4) with an upper intercept $^{207}$Pb/$^{206}$Pb$_i$ = 0.8603 ± 0.0033. The f$^{207}$% of apatite analyses in Group 1 range

from 10.14–99.91%, with a mean of 78.65%. The slight over-dispersion of the apatite regression line may reflect some variation in Pb closure temperature in these crystals[84].

**Orcadian basin apatite U–Pb results.** The second apatite U–Pb session yielded 138 analyses from samples CQ1 and AQ1. These data form three discordant mixing arrays between radiogenic and common Pb components on a Tera–Wasserburg plot.

An unanchored regression through Group 1 apatite ($n$ = 14) from the Cruaday sample (CQ1) yields a lower intercept of 473 ± 25 Ma (MSWD = 1.8) with an upper intercept of $^{207}$Pb/$^{206}$Pb$_i$ = 0.8497 ± 0.0128. The f$^{207}$% spans 38–99%, with a mean value of 85%.

Group 1 from the Spittal sample (AQ1), comprised of 109 analyses, yields a lower intercept equal to 466 ± 6 Ma (MSWD = 1.2). The upper $^{207}$Pb/$^{206}$Pb$_i$ is equal to 0.8745 ± 0.0038. f$^{207}$% values for this group range from 6–99%, with a mean value of 83%. A regression through Group 2 analyses ($n$ = 17) from the Spittal sample yields a lower intercept of 1,013 ± 35 Ma (MSWD = 1) and an upper intercept $^{207}$Pb/$^{206}$Pb$_i$ of 0.9038 ± 0.0101. f$^{207}$% values span 25–99%, with a mean of 76%. Combined U–Pb analyses from Groups 1 from CQ1 and AQ1 ($n$ = 123) yield a lower intercept equivalent to 466 ± 6 Ma (MSWD = 1.4) and an upper intercept $^{207}$Pb/$^{206}$Pb$_i$ of 0.8726 ± 0.0036, which is presented beneath the Orcadian Basin kernel density estimate in Fig. 4b.

**Apatite Lu–Hf methods.** Apatite grains were dated in thin-section by the in situ Lu–Hf method at the University of Adelaide, using a RESOlution-LR 193 nm excimer laser ablation system, coupled to an Agilent 8900 ICP–MS/MS[85,86]. A gas mixture of NH$_3$ in He was used in the mass spectrometer reaction cell to promote high-order Hf reaction products, while equivalent Lu and Yb reaction products were negligible. The mass-shifted (+82 amu) reaction products of $^{176+82}$Hf and $^{178+82}$Hf reached the highest sensitivity of the measurable range and were analysed free from isobaric interferences. $^{177}$Hf was calculated from $^{178}$Hf, assuming natural abundances. $^{175}$Lu was measured on mass as a proxy[85] for $^{176}$Lu. Laser ablation was conducted with a laser beam of 43 µm at 7.5 Hz repetition rate and a fluency of approximately 3.5 J cm$^{-2}$. The analysed isotopes (with dwell times in ms between brackets) are $^{27}$Al (2), $^{43}$Ca (2), $^{57}$Fe (2), $^{88}$Sr (2), $^{89+85}$Y (2), $^{90+83}$Zr (2), $^{140+15}$Ce (2), $^{146}$Nd (2), $^{147}$Sm (2), $^{172}$Yb (5), $^{175}$Lu (10), $^{175+82}$Lu (50), $^{176+82}$Hf (200) and $^{178+82}$Hf (150). Isotopes with short dwell times (<10 ms) were measured to confirm apatite chemistry and to monitor for inclusions. $^{175+82}$Lu was monitored for interferences on $^{176+82}$Hf.

Relevant isotope ratios were calculated in LADR[87] using NIST 610 as the primary reference material[88]. Subsequently, reference apatite OD-306[78] (1,597 ± 7 Ma) was used to correct the Lu–Hf isotope ratios for matrix-induced fractionation[86,89]. Reference apatites Bamble-1 (1,597 ± 5 Ma), HR-1 (344 ± 2 Ma) and Wallaroo (1,574 ± 6 Ma) were monitored for accuracy verification[85,86,90]. Measured Lu–Hf dates of 1,098 ± 7 Ma, 346.0 ± 3.7 Ma and 1,575 ± 12 Ma, respectively, are in agreement with published values. All reference materials have negligible initial Hf, and weighted mean Lu–Hf dates were calculated in IsoplotR[61] directly from the (matrix-corrected) $^{176}$Hf/$^{176}$Lu ratios.

For the Altar Stone apatites, which have variable $^{177}$Hf/$^{176}$Hf compositions, single-grain Lu–Hf dates were calculated by anchoring isochrons to an initial $^{177}$Hf/$^{176}$Hf composition[90] of 3.55 ± 0.05, which spans the entire range of initial $^{177}$Hf/$^{176}$Hf ratios of the terrestrial reservoir (for example, ref. 91). The reported uncertainties for the single-grain Lu–Hf dates are presented as 95% confidence intervals, and dates are displayed on a kernel density estimate plot.

**Apatite Lu–Hf results.** Forty-five apatite Lu–Hf analyses were obtained from 2010K.240. Those with radiogenic Lu ingrowth or lacking common Hf gave Lu–Hf ages, defining four coherent isochrons and age groups.

Group 1, defined by 16 grains, yields a Lu–Hf isochron with a lower intercept of 470 ± 28 Ma (MSWD = 0.16, $p(\chi^2)$ = 1). A second isochron through 5 analyses (Group 2) constitutes a lower intercept equivalent to

$604 \pm 38$ Ma (MSWD = 0.14, $p(\chi^2)$ = 0.94). Twelve apatite Lu–Hf analyses define Group 3 with a lower intercept of $1{,}123 \pm 42$ Ma (MSWD = 0.75, $p(\chi^2)$ = 0.68). Three grains constitute the oldest grouping, Group 4 at $1{,}526 \pm 186$ Ma (MSWD = 0.014, $p(\chi^2)$ = 0.91).

**Apatite trace elements methods.** A separate session of apatite trace element analysis was undertaken. Instrumentation and analytical set-up were identical to that described in 4.1. NIST 610 glass was the primary reference material for apatite trace element analyses. $^{43}$Ca was used as the internal reference isotope, assuming an apatite Ca concentration of 40 wt%. Secondary reference materials included NIST 612 and the BHVO–2g glasses[92]. Elemental abundances for secondary reference material were generally within 5–10% of accepted values. Apatite trace element data was examined using the Geochemical Data Toolkit[93].

**Apatite trace elements results.** One hundred and thirty-six apatite trace element analyses were obtained from as many grains. Geochemical classification schemes for apatite were used[29], and three compositional groupings (felsic, mafic-intermediate, and alkaline) were defined.

Felsic-classified apatite grains ($n$ = 83 (61% of analyses)) are defined by La/Nd of <0.6 and (La + Ce + Pr)/ΣREE (rare earth elements) of <0.5. The median values of felsic grains show a flat to slightly negative gradient on the chondrite-normalized REE plot from light to heavy REEs[94]. Felsic apatite's median europium anomaly (Eu/Eu*) is 0.59, a moderately negative signature.

Mafic-intermediate apatite[29] ($n$ = 48 (35% of grains)) are defined by (La + Ce + Pr)/ΣREE of 0.5–0.7 and a La/Nd of 0.5–1.5. In addition, apatite grains of this group typically exhibit a chondrite-normalized Ce/Yb of >5 and ΣREEs up to 1.25 wt%. Apatite grains classified as mafic-intermediate show a negative gradient on a chondrite-normalized REE plot from light to heavy REEs. The apatite grains of this group generally show the most enrichment in REEs compared to chondrite[94]. The median europium (Eu/Eu*) of mafic-intermediate apatite is 0.62, a moderately negative anomaly.

Lastly, alkaline apatite grains[29] ($n$ = 5 (4% of analyses)) are characterized by La/Nd > 1.5 and a (La + Ce + Pr)/ΣREE > 0.8. The median europium anomaly of this group is 0.45. This grouping also shows elevated chondrite-normalized Ce/Yb of >10 and >0.5 wt% for the ΣREEs.

## Reporting summary

Further information on research design is available in the Nature Portfolio Reporting Summary linked to this article.

## Data availability

The isotopic and chemical data supporting the findings of this study are available within the paper and its supplementary information files.

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

**Acknowledgements** Funding was provided by an Australian Research Council Discovery Project (DP200101881). Sample material was loaned from the Salisbury Museum and Amgueddfa Cymru–Museum Wales and sampled with permission. The authors thank A. Green for assistance in accessing the Salisbury Museum material; B. McDonald, N. Evans, K. Rankenburg and S. Gilbert for their help during isotopic analysis; and P. Sampaio for assistance with statistical analysis. Instruments in the John de Laeter Centre, Curtin University, were funded via AuScope, the Australian Education Investment Fund, the National Collaborative Research Infrastructure Strategy, and the Australian Government. R.E.B. acknowledges a Leverhulme Trust Emeritus Fellowship.

**Author contributions** A.J.I.C.: writing, original draft, formal analysis, investigation, visualization, project administration, conceptualization and methodology. C.L.K.: supervision,

resources, formal analysis, funding acquisition, writing, review and editing, conceptualization and methodology. R.E.B.: writing, review and editing, resources and conceptualization. N.J.G.P.: writing, review and editing, resources and conceptualization. S.G.: resources, formal analysis, funding acquisition, writing, review and editing, supervision, and methodology. R.A.I.: writing, review and editing.

**Competing interests** The authors declare no competing interests.

**Additional information**
**Correspondence and requests for materials** should be addressed to Anthony J. I. Clarke.

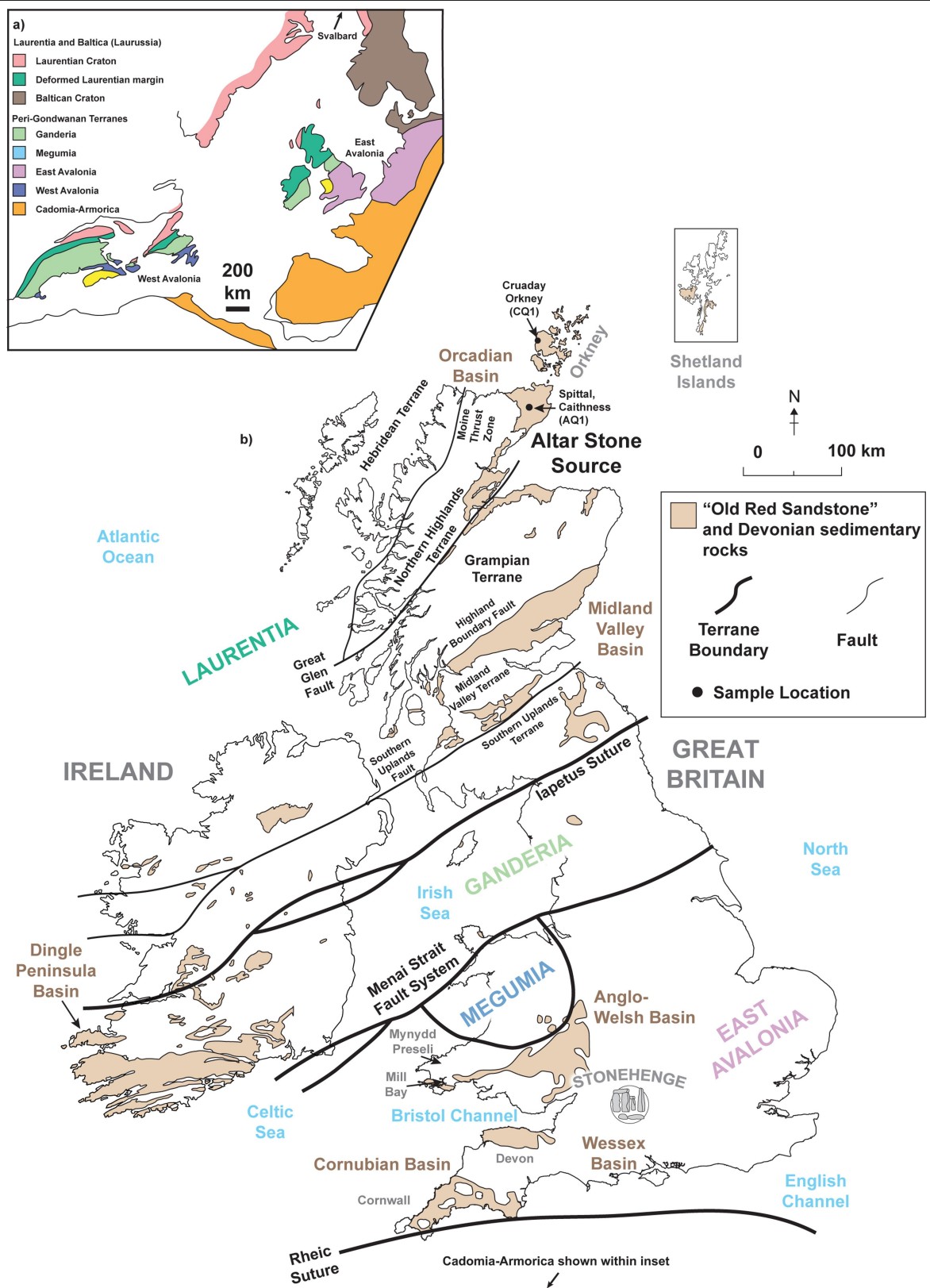

**Extended Data Fig. 1 | Geological maps of potential source terranes for the Altar Stone. a**, Schematic map of the North Atlantic region with the crystalline terranes in the Caledonian-Variscan orogens depicted prior to the opening of the North Atlantic, adapted after ref. 95. **b**, Schematic map of Britain and Ireland, showing outcrops of Old Red Sandstone, basement terranes, and major faults with reference to Stonehenge.

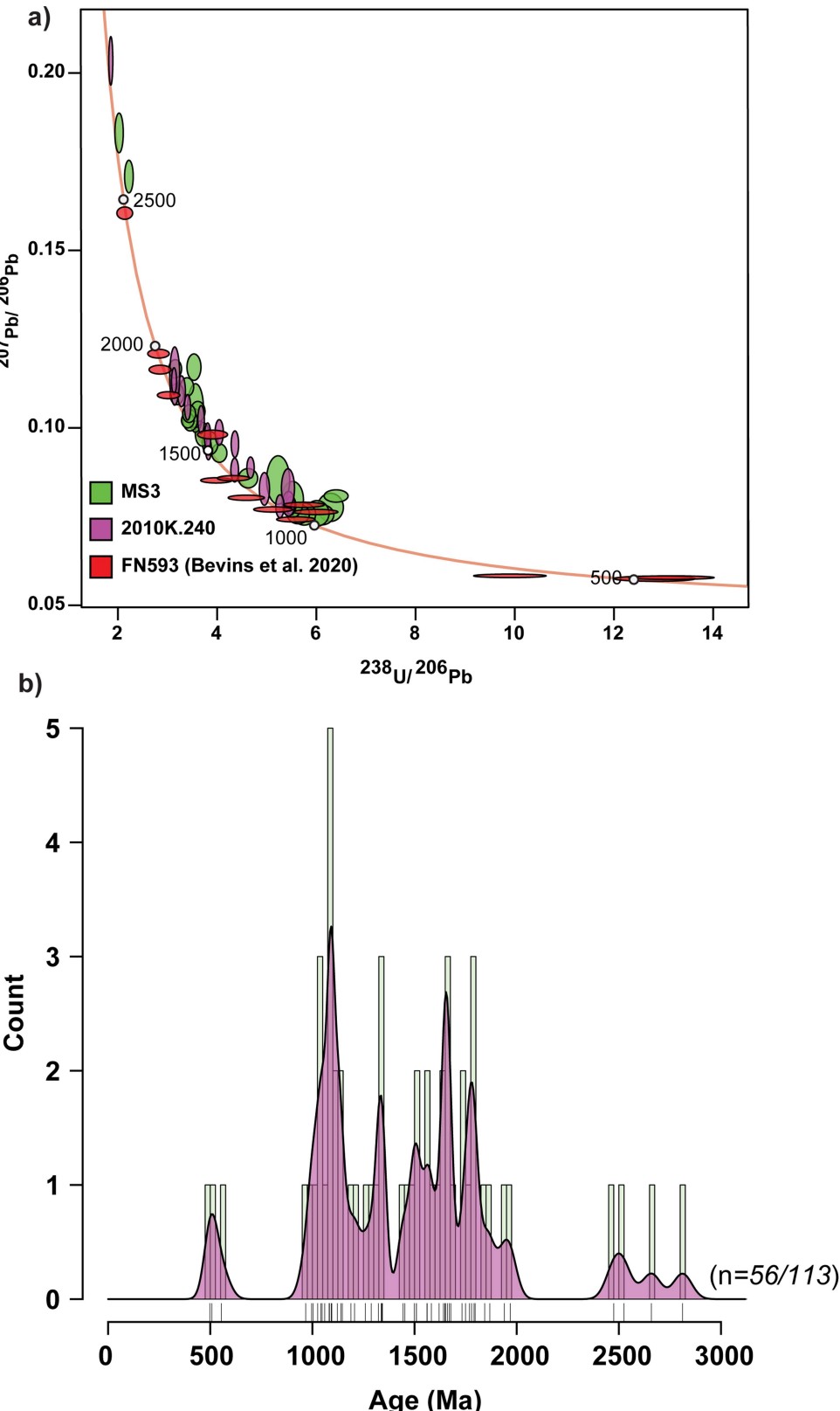

**Extended Data Fig. 2 | Altar Stone zircon U–Pb data. a**, Tera-Wasserburg plot for all concordant (≤10% discordant) zircon analyses reported from three samples of the Altar Stone. Discordance is defined using the concordia log % distance approach, and analytical ellipses are shown at the two-sigma uncertainty level. The ellipse colour denotes the sample. Replotted isotopic data for thin-section FN593 is from ref. 1. **b**, Kernel density estimate for concordia U–Pb ages of concordant zircon from the Altar Stone, using a kernel and histogram bandwidth of 50 Ma. Fifty-six concordant analyses are shown from 113 measurements. A rug plot is given below the kernel density estimate, marking the age of each measurement.

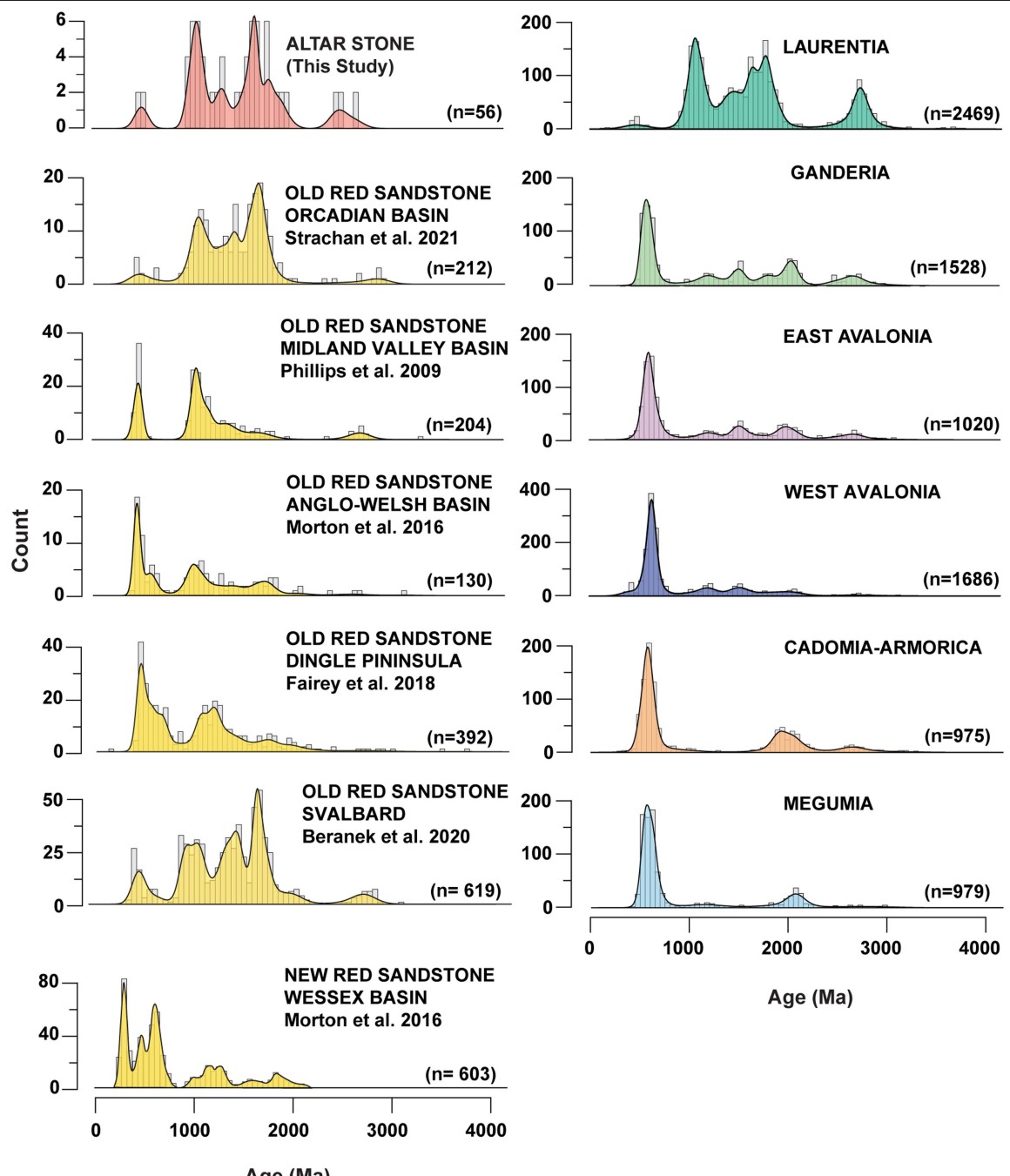

**Extended Data Fig. 3 | Comparative kernel density estimates of concordant zircon concordia ages from the Altar Stone, crystalline sources terranes, and comparative sedimentary rock successions.** Each plot uses a kernel and histogram bandwidth of 50 Ma. The zircon U–Pb geochronology source for each comparative dataset is shown with their respective kernel density estimate. Zircon age data for basement terranes (right side of the plot) was sourced from refs. 20,26.

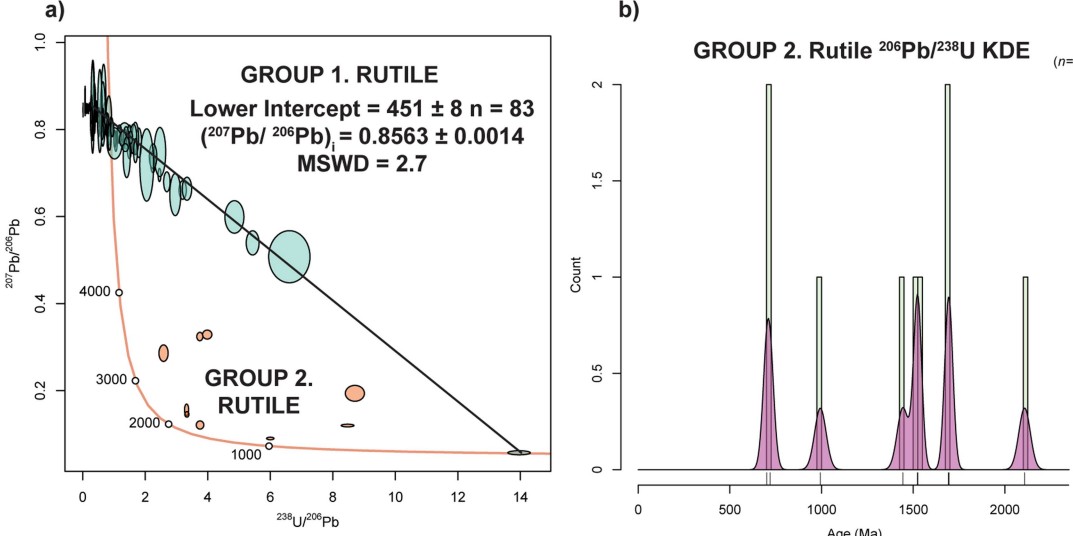

**a)**

**GROUP 1. RUTILE**
**Lower Intercept = 451 ± 8 n = 83**
$(^{207}Pb/^{206}Pb)_i = 0.8563 ± 0.0014$
**MSWD = 2.7**

**GROUP 2. RUTILE**

$^{207}Pb/^{206}Pb$

$^{238}U/^{206}Pb$

**b)**

**GROUP 2. Rutile $^{206}Pb/^{238}U$ KDE**  *(n=9)*

Count

Age (Ma)

**Extended Data Fig. 4 | Plots of rutile U–Pb ages. a**, Tera-Wasserburg plot of rutile U–Pb analyses from the Altar Stone (thin-section MS3). Isotopic data is shown at the two-sigma uncertainty level. **b**, Kernel density estimate for Group 2 rutile $^{207}Pb$ corrected $^{206}Pb/^{238}U$ ages, using a kernel and histogram bandwidth of 25 Ma. The rug plot below the kernel density estimate marks the age for each measurement.

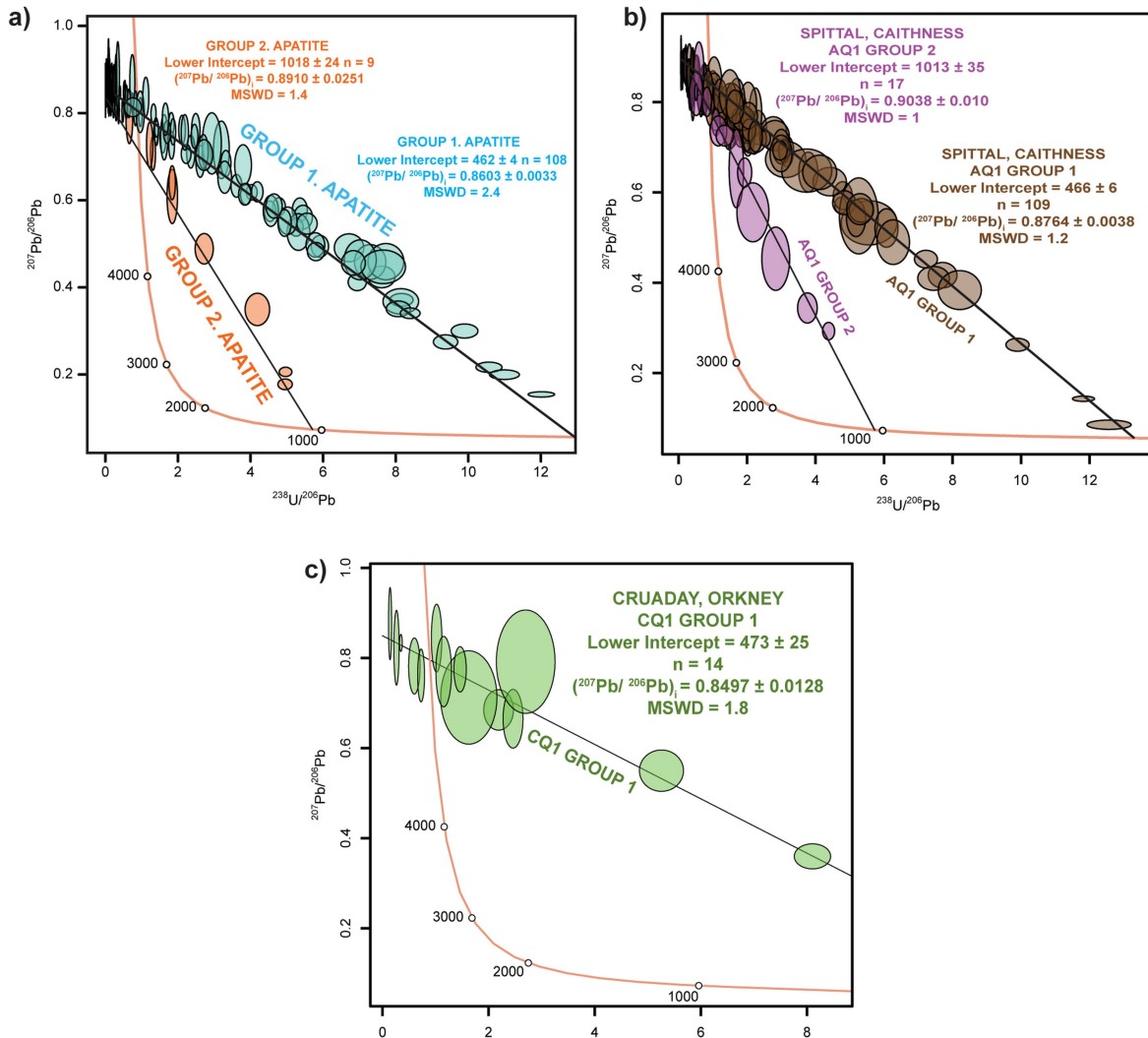

**Extended Data Fig. 5 | Apatite Tera-Wasserburg U–Pb plots for the Altar Stone and Orcadian Basin. a**, Altar Stone apatite U–Pb analyses from thin-section MS3. **b**, Orcadian Basin apatite U–Pb analyses from sample AQ1, Spittal, Caithness. **c**, Orcadian Basin apatite U–Pb analyses from sample CQ1, Cruaday, Orkney. All data are shown as ellipses at the two-sigma uncertainty level. Regressions through U–Pb data are unanchored.

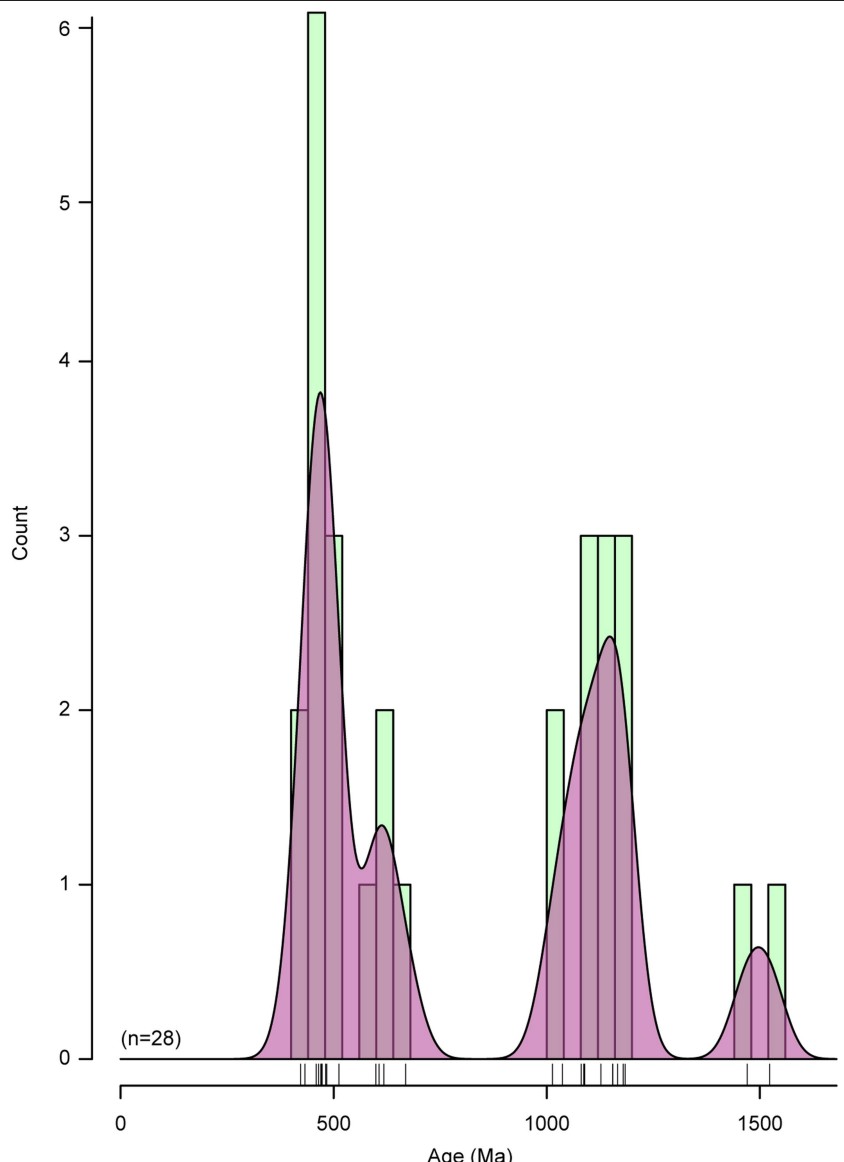

**Extended Data Fig. 6 | Combined kernel density estimate and histogram for apatite Lu–Hf single-grain ages from the Altar Stone.** Lu–Hf apparent ages from thin-section 2010K.240. Kernel and histogram bandwidth of 50 Ma. The rug plot below the kernel density estimate marks each calculated age. Single spot ages are calculated assuming an initial average terrestrial $^{177}Hf/^{176}Hf$ composition (see Methods).

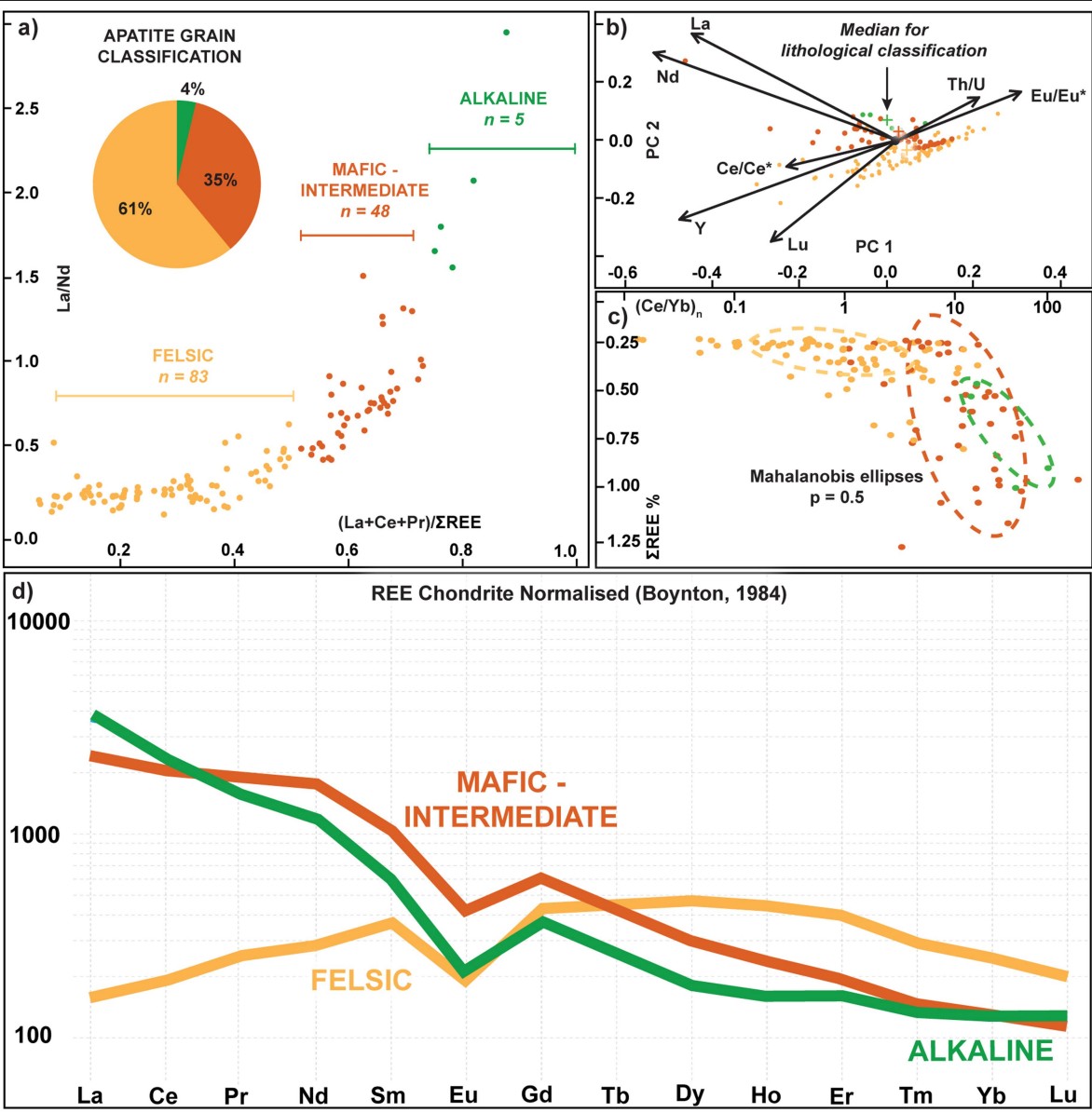

**Extended Data Fig. 7 | Apatite trace element classification plots for the Altar Stone thin-section MS3.** Colours for all plots follow the geochemical discrimination defined in A. **a**, Reference 29 classification plot for apatite with an inset pie chart depicting the compositional groupings based on these geochemical ratios. **b**, The principal component plot of geochemical data from apatite shows the main eigenvectors of geochemical dispersion, highlighting enhanced Nd and La in the distinguishing groups. Medians for each group are denoted with a cross. **c**, Plot of total rare earth elements (REE) (%) versus $(Ce/Yb)_n$ with Mahalanobis ellipses around compositional classification centroids. A P = 0.5 in Mahalanobis distance analysis represents a two-sided probability, indicating that 50% of the probability mass of the chi-squared distribution for that compositional grouping is contained within the ellipse. This probability is calculated based on the cumulative distribution function of the chi-squared distribution. **d**, Chondrite normalized REE plot of median apatite values for each defined apatite classification type.

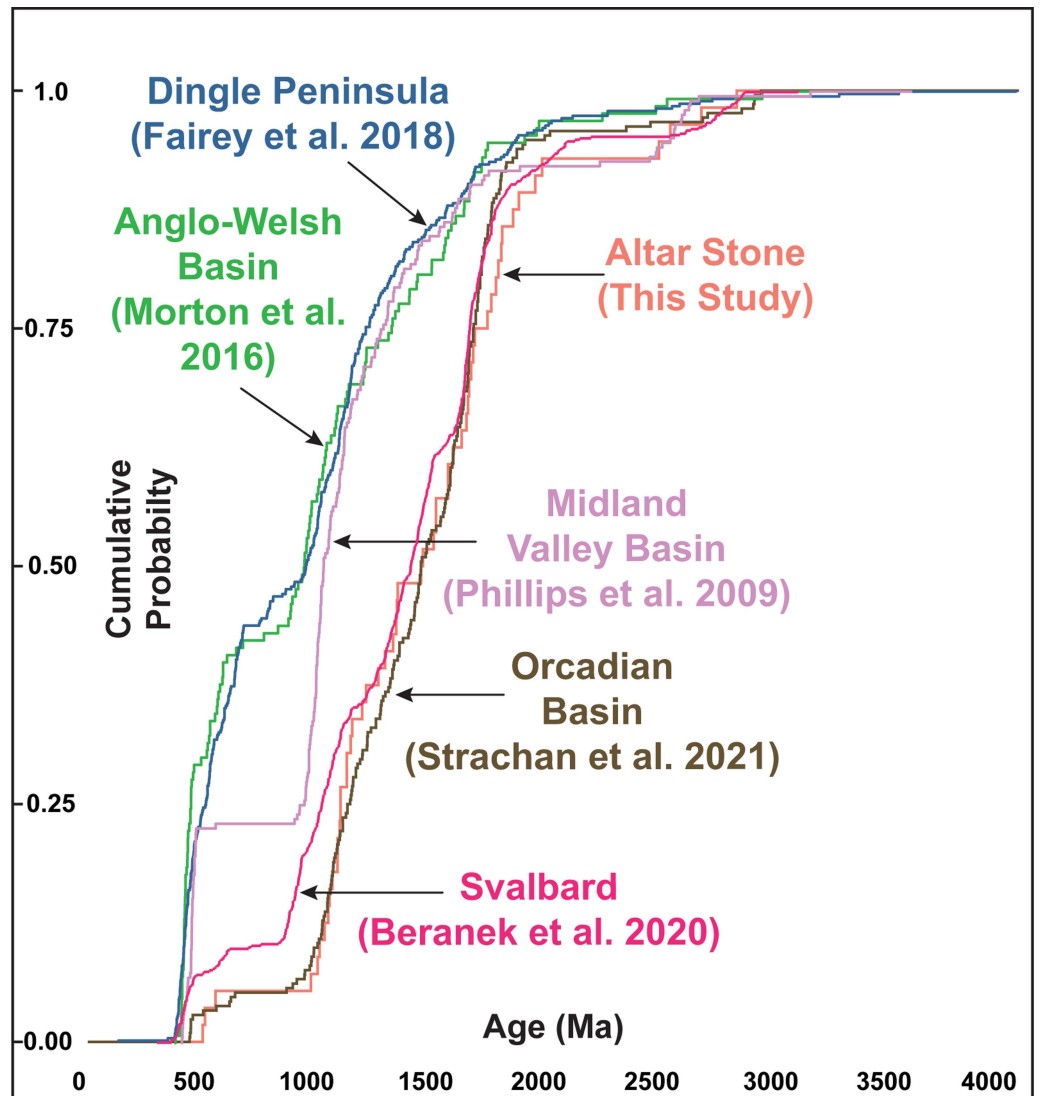

**Extended Data Fig. 8 | Cumulative probability density function plot.**
Cumulative probability density function plot of comparative Old Red Sandstone detrital zircon U–Pb datasets (concordant ages) versus the Altar Stone. Proximity between cumulative density probability lines implies similar detrital zircon age populations.

# Reporting Summary

Please do not complete any field with "not applicable" or n/a. Refer to the help text for what text to use if an item is not relevant to your study.
For final submission: please carefully check your responses for accuracy; you will not be able to make changes later.

## Statistics

For all statistical analyses, confirm that the following items are present in the figure legend, table legend, main text, or Methods section.

| n/a | Confirmed | |
|---|---|---|
| ☐ | ☒ | The exact sample size (*n*) for each experimental group/condition, given as a discrete number and unit of measurement |
| ☐ | ☒ | A statement on whether measurements were taken from distinct samples or whether the same sample was measured repeatedly |
| ☐ | ☒ | The statistical test(s) used AND whether they are one- or two-sided *Only common tests should be described solely by name; describe more complex techniques in the Methods section.* |
| ☐ | ☒ | A description of all covariates tested |
| ☐ | ☒ | A description of any assumptions or corrections, such as tests of normality and adjustment for multiple comparisons |
| ☐ | ☒ | A full description of the statistical parameters including central tendency (e.g. means) or other basic estimates (e.g. regression coefficient) AND variation (e.g. standard deviation) or associated estimates of uncertainty (e.g. confidence intervals) |
| ☐ | ☒ | For null hypothesis testing, the test statistic (e.g. *F*, *t*, *r*) with confidence intervals, effect sizes, degrees of freedom and *P* value noted *Give P values as exact values whenever suitable.* |
| ☒ | ☐ | For Bayesian analysis, information on the choice of priors and Markov chain Monte Carlo settings |
| ☒ | ☐ | For hierarchical and complex designs, identification of the appropriate level for tests and full reporting of outcomes |
| ☒ | ☐ | Estimates of effect sizes (e.g. Cohen's *d*, Pearson's *r*), indicating how they were calculated |

*Our web collection on statistics for biologists contains articles on many of the points above.*

## Software and code

Policy information about availability of computer code

| Data collection | Zircon, apatite and rutile U-Pb isotopic and trace element data were collected using an Agilent 8900 triple quadrupole mass spectrometer. Apatite Lu-Hf data was measured using an Agilent 8900 ICP-MS/MS. Further details of analytical set-up can be found in the Methods and Supplementary Information. |
|---|---|
| Data analysis | For zircon, apatite and rutile U-Pb data reductions Iolite Version 4 (Paton et al 2011) and Isoplot R Version 6.2 (Vermeesch 2018) software were used. For zircon and rutile U-Pb data reduction the "U-Pb Geochronology" Data Reduction Scheme (DRS) was used in Iolite 4. Apatite trace element data was reduced using the "Trace Element" DRS in Iolite 4. Multidimensional scaling plots for zircon age datasets were created using the MATLAB script of Nordsvan et al. (2020), where MATLAB Version 9.11.0 (R2021b Update 2) was used. Apatite U-Pb isotopic data was reduced using the VizualAge UcomPbine DRS of Iolite Version 4. Apatite Lu-Hf ratios were calculate using the LADR software, Version 1.1.7.0 (Build date: 2021-11-03) (Norris & Danyushevsky, 2018) and Isoplot R. |

For manuscripts utilizing custom algorithms or software that are central to the research but not yet described in published literature, software must be made available to editors and reviewers. We strongly encourage code deposition in a community repository (e.g. GitHub). See the Nature Portfolio guidelines for submitting code & software for further information.

## Data

Policy information about availability of data

All manuscripts must include a data availability statement. This statement should provide the following information, where applicable:
- Accession codes, unique identifiers, or web links for publicly available datasets
- A description of any restrictions on data availability
- For clinical datasets or third party data, please ensure that the statement adheres to our policy

> The authors declare that the isotopic and chemical data supporting the findings of this study are available within the paper and its supplementary information files (1-3).

## Research involving human participants, their data, or biological material

Policy information about studies with human participants or human data. See also policy information about sex, gender (identity/presentation), and sexual orientation and race, ethnicity and racism.

| | |
|---|---|
| Reporting on sex and gender | Not applicable |
| Reporting on race, ethnicity, or other socially relevant groupings | Not applicable |
| Population characteristics | Not applicable |
| Recruitment | Not applicable |
| Ethics oversight | Not applicable |

Note that full information on the approval of the study protocol must also be provided in the manuscript.

# Field-specific reporting

Please select the one below that is the best fit for your research. If you are not sure, read the appropriate sections before making your selection.

☒ Life sciences  ☐ Behavioural & social sciences  ☐ Ecological, evolutionary & environmental sciences

For a reference copy of the document with all sections, see nature.com/documents/nr-reporting-summary-flat.pdf

# Life sciences study design

All studies must disclose on these points even when the disclosure is negative.

| | |
|---|---|
| Sample size | Sample size for zircon, apatite and rutile data was dictated by the availability of sufficiently sized grains for laser-ablation within the thin-sections analysed. For compilations of zircon U–Pb ages from source terranes we used compilations by Fairey et al. (2018) and Stevens and Baykal (2021). For zircon U–Pb ages from Old Red Sandstone basins we calculated concordia ages based on reported isotopic data and used the concordia log distance % approach (≤ 10 % discordant) to filter ages. |
| Data exclusions | No data was excluded from the study. |
| Replication | All reported ages for secondary reference material are within two standard error of accepted values. Furthermore, we detail our analytical and instrument set-up within the methods. |
| Randomization | Randomization was not applicable to this study. We collected isotopic and chemical data from thin-sections of rock to determine the provenance of the Altar Stone - negating the need for randomization |
| Blinding | Blinding was not applicable to this study as we collected isotopic and chemical data for thin-sections of rock. |

# Reporting for specific materials, systems and methods

We require information from authors about some types of materials, experimental systems and methods used in many studies. Here, indicate whether each material, system or method listed is relevant to your study. If you are not sure if a list item applies to your research, read the appropriate section before selecting a response.

## Materials & experimental systems

| n/a | Involved in the study |
|-----|----------------------|
| ☒ | ☐ Antibodies |
| ☒ | ☐ Eukaryotic cell lines |
| ☐ | ☒ Palaeontology and archaeology |
| ☒ | ☐ Animals and other organisms |
| ☒ | ☐ Clinical data |
| ☒ | ☐ Dual use research of concern |
| ☒ | ☐ Plants |

## Methods

| n/a | Involved in the study |
|-----|----------------------|
| ☒ | ☐ ChIP-seq |
| ☒ | ☐ Flow cytometry |
| ☒ | ☐ MRI-based neuroimaging |

# Palaeontology and Archaeology

**Specimen provenance**

This work analysed two 30 μm polished thin sections of the Altar Stone, MS3 and 2010K.240. Both thin sections were prepared from rock fragments collected from Stonehenge, U.K. during archaeological digs of the Altar Stone during the 20th century and were loaned and sampled with permission of National Museum of Wales and Salisbury Museum. We also analysed two sections of Old Red Sandstone rock from the Orcadian Basin (CQ1 and AQ1). CQ1 is from Cruaday, Orkney (59°04'34.2" N, 3°18'54.6" W), and AQ1 is from near Spittal, Caithness (58°28'13.8" N, 3°27'33.6" W). Both Orcadian Basin geological rock samples were purchased from the UK company: Natural Wonders ltd Registered office 20 Grape Lane, Whitby, YO22 4BA, North Yorkshire, Company Registration Number 05427798.

**Specimen deposition**

MS3 and 2010K.240. were received on loan from National Museum of Wales and Salisbury Museum respectively.

**Dating methods**

New dates are provided. We report in-situ zircon, apatite and rutile U-Pb ages (calculated as single-spot concordia ages) from MS3 and 2010K.240. From 2010K.240 we report apatite Lu-Hf ages. For Orcadian Basin samples we report apatite U-Pb ages.

☒ Tick this box to confirm that the raw and calibrated dates are available in the paper or in Supplementary Information.

**Ethics oversight**

Thin sections MS3 and 2010K.240 were analysed with permission of the National Museum of Wales and Salisbury Museum respectively.

Note that full information on the approval of the study protocol must also be provided in the manuscript.

# Plants

**Seed stocks**

Not applicable

**Novel plant genotypes**

Not applicable

**Authentication**

Not applicable

