## [Peer Review File · Nature]

Manuscript Title: A Scottish Provenance for the Altar Stone of Stonehenge

Reviewer Comments & Author Rebuttals

Reviewer Reports on the Initial Version:

Referees' comments:

Referee #1 (Remarks to the Author):

Comments on Clarke et al. "A Scottish Provenance for the Altar Stone at Stonehenge".

This is an excellent, clearly written and provocative manuscript that is grounded in solid science and will be of broad interest. It presents the results of isotopic analyses of detrital mineral grains within the Altar Stone at the internationally known Neolithic World Heritage Site at Stonehenge within the UK. These analyses are used to present a compelling case that the Altar Stone was excavated from a source within NE Scotland, 750 km from Stonehenge. Such a distal source is much further afield than previously considered and thus the manuscript represents an original contribution and a significant advance in understanding. The conclusions imply marine transport from source to final resting place and a high degree of societal organization. The manuscript has no significant flaws that would prohibit publication. Given the status of Stonehenge, the findings will be of broad interest not only to archaeologists, anthropologists and geoscientists but also the general public. A high degree of media interest would be predicted. Statistical tests and analytical results are all correctly reported.

The following comments may assist the authors in fine-tuning their arguments:

- 1) Prior to line 73, surely a sentence or two is needed to summarise the depositional environments of the ORS **AND** likely directions of sediment transport for the different basins (the latter could be inserted as arrows onto Fig 4a). This feeds into subsequent sections of the paper.
- 2) The authors are spot-on in assessing that a Laurentian provenance is much more likely than any of the Gondwanan-derived terranes from south of the Iapetus Suture. However, perhaps it needs to be clarified that the bulk of transport of these detrital grains likely occurred in the Neoproterozoic rather than the Devonian? What is not considered at all is whether this detritus could instead have been derived, at least in part, from SW Baltica. It is generally accepted that it is very difficult to discriminate between Laurentia and Baltica as potential sediment sources. Baltica would have been relatively close to Scotland in any Devonian reconstruction (see McKellar et al. 2019 and Strachan et al. 2021 who both advocated some ORS derivation from Baltica) and incorporates extensive areas of relatively pristine Mesoproterozoic basement, although I accept that there is little Archaean present. Some nuance is required here.

- 3) Line 108, which 'detrital mineral bands'? First mention of these, more context required.
- 4) Line 123, the only felsic intrusion of this age in the Northern Highlands is the Glen Dessary syenite (c. 448 Ma). 470-440 Ma plutons are much more common in the Grampian Highlands which further restricts the potential source for the Altar Stone.
- 5) Fig 4a – I know it is going to make it bigger but Ireland should be added with the same degree of detail as for England/Wales/Scotland – otherwise discussion on the SW Ireland ORS will be somewhat obscure to international readers.

Rob Strachan
Portsmouth
23/01/24

Referee #2 (Remarks to the Author):

Review of 'A Scottish Provenance for the Altar Stone of Stonehenge'. I am happy to recommend this manuscript for publication, pending some minor revisions.

The authors present a logical and compelling argument for a Scottish origin of the Altar Stone, but, my reservation is that some of this will be inaccessible to non-geologists (and given the subject this will be read by a wider audience):

- the terminology could be adapted accordingly, and in some instances simplified. An example: in distinguishing different sources of the rutiles, the authors refer to 'Pinwarian' magmatism, but, this is never referred to again, and it might have been more helpful to quote the age range for this event. There are also some inconsistencies: some rocks are described at Group level, others to the Formation etc.

- the figures and extensive supplementary Extended data use terminology not found in the main manuscript e.g. figures 3 and 4 refer to the different crystalline terranes of Britain – Ganderia, East Avalonia etc.; these are not referred to in the main text

- as the manuscript contains so much data – it is natural for the reader to look at the figures for reference. The lack of continuity between the main text and figures makes this more complicated than it should be (please also check continuity in the Extended Data tables, and figures 3 and 4)

- the manuscript repeatedly refers to the detrital record for the Midland Valley Basin, making a distinction between this and the record for Orcadian Basin. This classification is used in figure 3, but not figure 4. In figure 3b, 'Detrital zircon ages reported from Orcadian Basin ORS provide the closest match to the Altar Stone 23 (Figure 3b)' – but, the cumulative probability plot does not appear to distinguish between the ORS? In figure 4, the detrital record for the Midland Valley Basin is shown within the larger Laurentia dataset – so, does not substantiate sentences such as '[...] grains as young as 402 ± 5 Ma 34 from the northern ORS in the Midland Valley Basin, further differentiates this basin from the Altar Stone'

Referee #3 (Remarks to the Author):

This paper presents isotopic evidence for a Scottish provenance of the mysterious 'Altar Stone' at Stonehenge. The authors posit that the megalith was transported to Wiltshire by sea. I find the evidence for a Scottish provenance quite strong, although I do have some suggestions to further strengthen it. I am less convinced about the marine transport mechanism, but should add that I am not an archaeologist, so my opinion on this aspect of the study is that of an interested layman. I have three comments on the paper.

1. Possible addition of further evidence

Stonehenge is one of the most famous archaeological sites in the world. If published, the results of this study will make newspaper headlines all over the world. It is therefore important that the evidence is as strong as possible. I believe that the addition of some more data could make the case for a Scottish provenance even stronger than it is now.

Stonehenge in general, and the Altar Stone in particular, are incredibly precious. It is amazing that the research team has been able to obtain two thin sections from it for in-situ geochronology. They have squeezed as much geochronological information from these two thin sections as possible, including U-Pb measurements on zircon, rutile and apatite, and Lu-Hf measurements on apatite. So far, so good.

The argument for a Scottish provenance rests nearly entirely on 56 concordant zircon U-Pb dates, which include data from a second fragment of Altar Stone that was analysed by another group. The U-Pb data have been statistically compared to published U-Pb age spectra from other parts of Britain, Ireland and Brittany, using the Kolmogorov-Smirnov test and multidimensional scaling. The other geochronological data are presented as age spectra and concordia diagrams, but are not directly compared (neither statistically, nor qualitatively) with other samples from the various provenance candidates.

Whereas it is extremely difficult to obtain thin sections from Stonehenge, it is much easier to get thin sections from Wales and Scotland. So I am a little puzzled why the research team hasn't compared the in-situ rutile, apatite and Lu-Hf results from the Altar Stone with similar data from Old Red Sandstone samples elsewhere, including the Preseli Hills and the Grampian mountains. If the Australian authors are unable to visit these places themselves, then they could perhaps obtain such samples from the vast BGS rock collection. I feel that it would significantly strengthen their paper.

2. Transport mechanism

The authors propose that the Altar Stone was transported from Scotland to Wiltshire by sea. This would have profound implications for our understanding of Neolithic British society. Extraordinary claims require extraordinary evidence. So it is only right that we inspect the author's interpretation with a sceptical eye.

Suppose that the authors were correct, and that neolithic Britons had the technology to move a six-tonne block of sandstone from the Scottish highlands to southern England. Then this would beg the question why these people would go through so much trouble for such an ordinary type of rock? Why would they travel all the way to Scotland to obtain a plain looking block of Old Red Sandstone, when very similar looking rocks can be found just around the corner in Wales? Why not go for a more exotic rock type, such as a vesicular basalt? Marine transport of Old Red Sandstone from Scotland raises more questions than answers.

I am not an archaeologist but I do know that many prehistoric monuments (dolmen) in low lying areas are constructed from glacial erratics. So when I saw the evidence for a Scottish provenance of the Stonehenge bluestones, I thought that glacial transport would be far more likely than marine transport. The authors dismiss the glacial hypothesis in two sentences (lines 191-195):

“Some postulate a glacial transport mechanism for the Mynydd Preseli (Figure 4a) bluestones to Salisbury Plain. However, such transport for the Altar Stone is difficult to reconcile with ice-sheet reconstructions that show the northwards movement of glaciers (and erratics) from the Grampian Terrane towards the Orcadian Basin during the Last Glacial Maximum and, indeed, previous glaciations during the Pleistocene”

I did a quick literature search with the keywords “provenance of glacial erratics in Britain”, navigated to the first result (Williams-Thorpe et al., 1999) and found a map (see attached) with highly complex ice flow paths that do not rule out a Scottish provenance for Wiltshire erratics.

It may be so that a Scottish provenance is unlikely for any given erratic. However, prehistoric Britain contained hundreds of thousands of erratics (far more than today). If you multiply a small probability with a large number of erratics, then an unlikely origin may become quite likely. It seems plausible to me that prehistoric people would choose the rarest erratics to build their monuments. So they may have 'cherry picked' their bluestones for rarity, unwittingly creating the puzzle that Stonehenge presents to scientists today.

3. Statistical issues

According to lines 81-85 of the manuscript:

“Statistical comparisons between crystalline basement terranes, ORS, and the Altar Stone, made using a Kolmogorov-Smirnov (KS) test, indicate that at 95% confidence, no distinction in provenance is evident between Altar Stone detrital zircon U–Pb ages and those from Laurentian basement; that is we accept the hypothesis that both samples are from the same age distribution (P-value >0.05) (Figure 3a).”

Here the authors misrepresent the concept of statistical hypothesis testing. Failure to reject a null hypothesis does NOT mean that said null hypothesis has been accepted. The outcome of a statistical hypothesis test (such as KS) depends on two things: (1) the degree to which the null hypothesis is false, and (2) sample size. The second point is evident from Table 3 of the paper. For example, the D-value of “Dingle Peninsula” and “Anglo-Welsh Basin” is 0.10. This is exactly the same as the D-value for “Orcadian Basin” and “Altar Stone”. However, their p-values are different, at 0.33 and 0.00, respectively. The reason for this difference is that the “Altar Stone” sample is much smaller than the “Dingle Peninsula” and “Anglo-Welsh Basin” datasets.

As a second example, the D-value for “Laurentia” and “Altar Stone” is 0.22, which is higher than the D-value of 0.12 for “Ganderia” and “East Avalonia”. However, their p-value is lower. In other words, the D-values suggest that “Ganderia” and “East Avalonia” are more similar to each other than “Laurentia” and “Altar Stone”, but the p-values suggest the opposite!

An extended discussion of this phenomenon is provided by Vermeesch (2018), who also points out that this problem undermines the validity of Satkoski et al. (2013)’s ad-hoc Likeness measure of dissimilarity (which is mentioned on line 488 of the manuscript and can be removed without consequence).

Other comments:

Line 464: “smallest” should be “largest”

Line 497: "detectors" should be "mass analyser"
Lines 429 and 530: "Lauren" should be "Laurin"

[REDACTED]

Author Rebuttals to Initial Comments:

Reviewer 1	Reply
This is an excellent, clearly written and provocative manuscript that is grounded in solid science and will be of broad interest. It presents the results of isotopic analyses of detrital mineral grains within the Altar Stone at the internationally known Neolithic World Heritage Site at Stonehenge within the UK. These analyses are used to present a compelling case that the Altar Stone was excavated from a source within NE Scotland, 750 km from Stonehenge. Such a distal source is much further afield than previously considered and thus the manuscript represents an original contribution and a significant advance in understanding. The conclusions imply marine transport from source to final resting place and a high degree of societal organisation. The manuscript has no significant flaws that would prohibit publication. Given the status of Stonehenge, the findings will be of broad interest not only to archaeologists, anthropologists and geoscientists but also the general public. A high degree of media interest would be predicted. Statistical tests and analytical results are all correctly reported.	Thank you for your support of our work. We appreciate the insightful comments, which have helped strengthen our manuscript. Below, we address your comments, indicating our changes in bold.
Prior to line 73, surely a sentence or two is needed to summarise the depositional environments of the ORS AND likely directions of sediment transport for the different basins (the latter could be inserted as arrows onto Fig 4a). This feeds into subsequent sections of the paper.	Agreed, some further background on the ORS could be a helpful addition. “Previous petrographic work on the Altar Stone has implied an association to the Old Red Sandstone (ORS)⁹⁻¹¹. The ORS is a late Silurian to Devonian sedimentary rock assemblage that crops out widely throughout Great Britain and Ireland (Extended Data Figure 1). ORS lithologies are dominated by terrestrial siliciclastics sedimentary rocks deposited in continental fluvial, lacustrine, and aeolian

	environments¹². Each ORS basin reflects local subsidence and sediment infill and thus contains proximal crystalline signatures^{12,13}.”
The authors are spot-on in assessing that a Laurentian provenance is much more likely than any of the Gondwanan-derived terranes from south of the Iapetus Suture. However, perhaps it needs to be clarified that the bulk of transport of these detrital grains likely occurred in the Neoproterozoic rather than the Devonian? What is not considered at all is whether this detritus could instead have been derived, at least in part, from SW Baltica. It is generally accepted that it is very difficult to discriminate between Laurentia and Baltica as potential sediment sources. Baltica would have been relatively close to Scotland in any Devonian reconstruction (see McKellar et al. 2019 and Strachan et al. 2021 who both advocated some ORS derivation from Baltica) and incorporates extensive areas of relatively pristine Mesoproterozoic basement, although I accept that there is little Archaean present. Some nuance is required here.	We agree it is difficult to distinguish a Baltican versus Laurentian source, although both ultimately support an Orcadian Basin provenance. We enhance the text around this aspect: “During the Palaeozoic, the Orcadian Basin was situated between Laurentia and Baltica on the Laurussian palaeocontinent^{13,35}. Correlations between detrital zircon age components imply that both Laurentia and Baltica supplied sediment into the Orcadian Basin^{24,35}. Detrital grains >900 Ma within the Altar Stone are consistent with sediment recycling from intermediary Neoproterozoic supracrustal successions (e.g. Dalradian Supergroup) within the Grampian Terrane but also from the Särvi and Sparagmite successions of Baltica^{24,35}. At ca. 470 Ma, the Grampian Terrane began to denude²⁷. Subsequently, first-cycle detritus, such as that represented by Group 1 apatite and rutile, was shed towards the Orcadian Basin from the southeast²⁴.” “Thus, the resistive mineral cargo in the Altar Stone represents a complex mix of first and multi-cycle grains from multiple sources. Regardless of total input from Baltica versus Laurentia into the Orcadian Basin, crystalline terranes north of the Iapetus Suture (Figure 4a) have distinct age components that match the Altar Stone in contrast to Gondwanan-derived terranes to the south.”
Line 108, which ‘detrital mineral bands’? First mention of these, more context required.	Ok, fair point. We have clarified this description by adding the following text: “Throughout the Altar Stone are sub-planar 100 – 200 µm bands of concentrated heavy resistive

	minerals. These resistive minerals are interpreted to be magmatic in origin, given internal textures (oscillatory zonation), lack of mineral overgrowths (in all dated minerals) (Figure 2) and the igneous apatite trace element signatures (Extended Data Figure 7; Supplementary Information 3) ²⁶. Moreover, there is a general absence of detrital metamorphic zircon grains, further supporting a magmatic origin for these grains.”
Line 123, the only felsic intrusion of this age in the Northern Highlands is the Glen Dessary syenite (c. 448 Ma). 470-440 Ma plutons are much more common in the Grampian Highlands which further restricts the potential source for the Altar Stone.	Thank you for this insight. The text now reads: “The alkaline to calc-alkaline suites in these terranes are volumetrically small, consistent with the scarcity of alkaline apatite grains within the Altar Stone (Extended Data 7). Indeed, the 448 ± 3 Ma Glen Dessary syenite is the only age-appropriate felsic-alkaline pluton in the Northern Highlands Terrane ²⁹.”
Fig 4a – I know it is going to make it bigger but Ireland should be added with the same degree of detail as for England/Wales/Scotland – otherwise discussion on the SW Ireland ORS will be somewhat obscure to international readers.	We are restricted by formatting requirements and are already at the line and font size limit for Figure 4. However, for the interested reader, we have added a map of ORS in Britain and Ireland as Extended Data Figure 1. Please see below:

Scale Sutures Cadomian/América shows within inset

Reviewer 2	Reply
I am happy to recommend this manuscript for publication, pending some minor revisions.	Thank you for the helpful suggestions. Please see our replies to your comments below. We indicate changes to the manuscript in bold.
The authors present a logical and compelling argument for a Scottish origin of the Altar Stone, but, my reservation is that some of this will be inaccessible to non-geologists (and given the subject this will be read by a wider audience): the terminology could be adapted accordingly, and in some instances simplified. An example: in distinguishing different sources of the rutiles, the authors refer to ‘Pinwarian’ magmatism, but, this is never referred to again, and it might have been more helpful to quote the age range for this event.	We have adapted terminology for accessibility. Tectonomagmatic events are now also provided with age ranges. "Detrital zircon age components, defined by concordant analyses from ≥ 4 grains in the Altar Stone, include maxima at 1047, 1091, 1577, 1663, and 1790 Ma (Extended Data Figure 2), corresponding to known tectonomagmatic events and sources within Laurentia and Baltica, including the Grenville (1095 – 980 Ma), Labrador (1690 – 1590 Ma), Gothian (1660 – 1520 Ma), and Svecokarellian (1920 – 1770 Ma) orogenies²⁴. Laurentian terranes are crystalline lithologies north of the Iapetus Suture Zone (which marks the collision zone between Laurentia and Avalonia) and include the Southern Uplands, Midland Valley, Grampian, Northern Highlands, and Hebridean Terranes (Figure 4a). Together, these terranes preserve a Proterozoic to Archaean record of zircon production²³, distinct from the southern Gondwanan-derived terranes of Britain (Figure 4a; Extended Data Figure 3)^{19,25}." "Age data from Altar Stone rutile grains also point towards an ultimate Laurentian source with several discrete age components (Extended Data Figure 4; Supplementary Information 1). Group 2 rutile U–Pb analyses from the Altar Stone include Proterozoic ages from 1724 – 591 Ma, with three grains constituting an age peak at 1607 Ma, overlapping with Laurentian magmatism, including the Labrador and Pinwarian (1690 – 1380 Ma) orogenies²³."

	We have also included a map of Britain and Ireland as Extended Data Figure 1 (see above), which helps place terranes and ORS basins in a broader geographic context.
There are also some inconsistencies: some rocks are described at Group level, others to the Formation etc.	Thanks for this point. Yes, we variably refer to the “Cosheston Subgroup” or “Senni Formation”, but this is within the context of previous investigations of the Altar Stone’s provenance (within the Anglo-Welsh Basin). However, we endeavour to start with an overview when needing to convey variable levels of lithological detail. Please see the modified text below: An early proposed source for the Altar Stone from Mill Bay, Pembrokeshire (Cosheston Subgroup of the Anglo-Welsh ORS Basin), close to the Mynydd Preseli source of the doleritic and rhyolitic bluestones, strongly influenced the notion of a sea transport route via the Bristol Channel ¹¹.
The figures and extensive supplementary Extended data use terminology not found in the main manuscript e.g. figures 3 and 4 refer to the different crystalline terranes of Britain – Ganderia, East Avalonia etc.; these are not referred to in the main text As the manuscript contains so much data – it is natural for the reader to look at the figures for reference. The lack of continuity between the main text and figures makes this more complicated than it should be (please also check continuity in the Extended Data tables, and figures 3 and 4)	Thanks for highlighting inconsistencies between Figures 3 and 4 and the manuscript. The text now reads: “The crystalline basement terranes of Great Britain and Ireland, from north to south, are Laurentia, Ganderia, Megumia, and East Avalonia (Figure 4a; Extended Data Figure 1). Cadomia-Armorica is south of the Rheic Suture and encompasses part of western Europe, including northern France and Spain. East Avalonia, Megumia, and Ganderia are partly separated by the Menai Strait Fault System (Figure 4a). Each terrane has discrete age components, which have imparted palaeogeographic information into overlying sedimentary basins ^{12,13,22}. Laurentia was a

	palaeocontinent that collided with Baltica and Avalonia (a peri-Gondwanan microcontinent) during the early Palaeozoic Caledonian Orogeny to form Laurussia^{13,23}. West Avalonia is a terrane that includes parts of eastern Canada and comprised the western margin of Avalonia (Extended Data Figure 1).” Other features from Figure 4 that were excluded from the manuscript included the Hebridean Terrane, Moine Thrust Zone, and Southern Uplands Fault. Please see their incorporation below: “In Scotland, ORS predominantly crops out in the Midland Valley and the Orcadian Basin (Figure 4a). The Midland Valley Basin is bound between the Highland Boundary Fault and Iapetus Suture and is located within the Midland Valley and Southern Uplands Terrane (Figure 4a). Throughout Midland Valley ORS stratigraphy, detrital zircon age spectra broadly show a bimodal age distribution between Lower Palaeozoic and Mesoproterozoic components^{34,35} (Extended Data Figure 3). Indeed, throughout 9 km of ORS stratigraphy in the Midland Valley Basin, and across the Southern Uplands Fault, no major changes in provenance are recognised³⁵ (Figure 4).” “The Wessex Basin, containing NRS, has characteristic detrital zircon age components, specifically, Carboniferous to Permian age zircon grains, unlike the Altar Stone (Extended Data Figure 3)^{1,22,25,32,33}.”
The manuscript repeatedly refers to the detrital record for the Midland Valley Basin, making a distinction between this and the record for Orcadian Basin. This classification is used in figure 3, but not figure 4. In figure 3b, ‘Detrital zircon ages reported from Orcadian Basin ORS provide the closest match to the Altar Stone 23 (Figure 3b)’ – but, the cumulative	Although geographically close, the Orcadian and Midland Valley Basins contain distinct detrital zircon age cargos. Hence, we wish to highlight this important distinction. Figure 3a is a multi-dimensional scaling plot of all comparative datasets at terrane and basin levels.

probability plot does not appear to distinguish between the ORSs?

Figure 3b is a cumulative probability plot of zircon ages from the Altar Stone, terranes, and the Orcadian Basin.

For clarity, we chose to omit other basins in Figure 3b. Nonetheless, a basin-level cumulative probability plot (at this more granular scale) is now provided for the interested reader as Extended Data Figure 8

In figure 4, the detrital record for the Midland Valley Basin is shown within the larger Laurentia dataset – so, does not substantiate sentences such as ‘[...] grains as young as 402 ± 5 Ma³⁴ from the northern ORS in the Midland Valley Basin, further differentiates this basin from the Altar Stone

Figure 4b shows a KDE of Laurentian zircon ages only. We included an Orcadian Basin KDE due to its striking similarity to Laurentia and the Altar Stone.

Our Extended Data now, however, now includes a KDE of Midland Valley Basin ages.

For reader clarity, we have edited the captions for Figures 3 and 4:

“Figure 3. Detrital zircon U–Pb ages from the Altar Stone and crystalline terranes of Britain and Ireland. a) Multi-dimensional scaling plot of concordant zircon U–Pb ages from the Altar Stone and comparative datasets, shown at 95% confidence⁵¹. b) Cumulative probability plot of zircon U–Pb ages from crystalline terranes, the Orcadian Basin and the Altar Stone. **For a**

cumulative probability plot of all basins, see Extended Data Figure 8.”

“Figure 4. a) Schematic map of the Britain Isles showing outcrops of Old Red Sandstone, basement terranes and major faults. Potential Caledonian source plutons are colour-coded based on age ²⁷. b) Kernel density estimate diagrams displaying the zircon U–Pb (red histogram), apatite Lu–Hf age (dashed line) spectra from the Altar Stone, the Orcadian Basin ²⁴ and plausible crystalline source terranes. The mid-Ordovician age component (443 – 466 Ma) defined by Population 1 apatite and Population 2 rutile U–Pb analyses (with uncertainty) is shown below the Altar Stone kernel density estimate. **Extended Data Figure 3 contains kernel density estimates of Old Red Sandstone and New Red Sandstone age datasets.”**

Reviewer 3	Reply
This paper presents isotopic evidence for a Scottish provenance of the mysterious 'Altar Stone' at Stonehenge. The authors posit that the megalith was transported to Wiltshire by sea. I find the evidence for a Scottish provenance quite strong, although I do have some suggestions to further strengthen it. I am less convinced about the marine transport mechanism, but should add that I am not an archaeologist, so my opinion on this aspect of the study is that of an interested layman. I have three comments on the paper. Stonehenge is one of the most famous archaeological sites in the world. If published, the results of this study will make newspaper headlines all over the world. It is therefore important that the evidence is as strong as possible. I believe that the addition of some more data could make the case for a Scottish provenance even stronger than it is now. Stonehenge in general, and the Altar Stone in particular, are incredibly precious. It is amazing that the research team has been able to obtain two thin sections from it for in-situ geochronology. They have squeezed as much geochronological information from these two thin sections as possible, including U-Pb measurements on zircon, rutile and apatite, and Lu-Hf measurements on apatite. So far, so good.	Thank you for your support of our work. We agree that the evidence for a Scottish provenance is compelling. Please see our responses to your comments below. We mark changes to the manuscript below in bold.

The argument for a Scottish provenance rests nearly entirely on 56 concordant zircon U-Pb dates, which include data from a second fragment of Altar Stone that was analysed by another group. The U-Pb data have been statistically compared to published U-Pb age spectra from other parts of Britain, Ireland and Brittany, using the Kolmogorov-Smirnov test and multi-dimensional scaling.

The other geochronological data are presented as age spectra and concordia diagrams, but are not directly compared (neither statistically, nor qualitatively) with other samples from the various provenance candidates.

Whereas it is extremely difficult to obtain thin sections from Stonehenge, it is much easier to get thin sections from Wales and Scotland. So I am a little puzzled why the research team hasn't compared the in-situ rutile, apatite and Lu-Hf results from the Altar Stone with similar data from Old Red Sandstone samples elsewhere, including the Preseli Hills and the Grampian mountains. If the Australian authors are unable to visit these places themselves, then they could perhaps obtain such samples from the vast BGS rock collection. I feel that it would significantly strengthen their paper.

You are right; zircon U-Pb analyses support our interpretation of a Scottish provenance for the Altar Stone. However, we do compare our apatite U-Pb dates with published data where available. For example, Fairey et al. (2018) report apatite U-Pb dates from Irish ORS. Late-Caledonian apatite in the Dingle Peninsula ORS, younger than the Altar Stone apatite, is a crucial difference and helps eliminate these successions as potential sources.

To further enhance this work, Orcadian Basin ORS apatite ages have been acquired from two sites in the Orcadian Basin in northeast Scotland: Achanarras and Cruaday in Caithness and Orkney, respectively. Both Orcadian samples reveal an Ordovician detrital apatite component consistent with early Caledonian magmatism; such a signature is also seen within the Altar Stone and **not** within the southern ORS.

Below is a Tera-Wasserburg plot for apatite data. Green ellipses are analysed using Altar Stone. The red ellipses are the Orcadian Basin. Apatite within the Altar Stone and Orcadian Basin are coeval.

The authors propose that the Altar Stone was transported from Scotland to Wiltshire by sea. This would have profound implications for our understanding of Neolithic British society. Extraordinary claims require extraordinary evidence. So it is only right that we inspect the author's interpretation with a sceptical eye.

Suppose that the authors were correct, and that neolithic Britons had the technology to move a six-tonne block of sandstone from the Scottish highlands to southern England. Then this would beg the question why these people would go through so much trouble for such an ordinary type of rock? Why would they travel all the way to Scotland to obtain a plain looking block of Old Red Sandstone, when very similar looking rocks can be found just around the corner in Wales? Why not go for a more exotic rock type, such as a vesicular basalt? Marine transport of Old Red Sandstone from Scotland raises more questions than answers.

Thank you for your insight. The robust provenance results based on analysis of resistive minerals allow a reasoned discussion on the transport mechanism. We articulate our logic for favouring anthropogenic versus glacial transport for the Altar Stone here and in the text. We apologise for the length of this response, but it is an aspect we have considered in detail.

1. The mechanics of glacial flow during the Last Glacial Maximum and previous glaciations during the Pleistocene (Clark et al. 2022) are incompatible with the southwards transport of the Altar Stone within ice. Recent comprehensive models of ice-flow patterns for Britain show the northwards movement of ice from the Grampian Mountains to the Orcadian Basin (Clark et al. 2022; Hughes et al. 2014). Submerged terminal moraines north of mainland Scotland, north-south orientated crag and tails, eskers, and drumlins in NE Scotland demonstrate northwards not southwards ice flow (Clark et al. 2022).
2. There is no evidence of ice movement on the Salisbury Plain and southern central Britain (McMillian et al. 2005; Gibbard & Clark, 2011;

I am not an archaeologist but I do know that many prehistoric monuments (dolmen) in low lying areas are constructed from glacial erratics. So when I saw the evidence for a Scottish provenance of the Stonehenge bluestones, I thought that glacial transport would be far more likely than marine transport. The authors dismiss the glacial hypothesis in two sentences (lines 191-195).

“Some postulate a glacial transport mechanism for the Mynydd Preseli (Figure 4a) bluestones to Salisbury Plain. However, such transport for the Altar Stone is difficult to reconcile with ice-sheet reconstructions that show the northwards movement of glaciers (and erratics) from the Grampian Terrane towards the Orcadian Basin during the Last Glacial Maximum and, indeed, previous glaciations during the Pleistocene”

I did a quick literature search with the keywords “provenance of glacial erratics in Britain”, navigated to the first result (Williams-Thorpe et al., 1999) and found a map (see attached) with highly complex ice flow paths that do not rule out a

Clark et al. 2012), and no erratics or glacially deposited sediments have been found at Stonehenge (Green, 1997). None of the megaliths at Stonehenge show evidence of ice transport, such as glacial striations (Bevins et al. 2023).

Figure 1 below shows a model of ice flow evolution throughout the Last Glacial Maximum. The Grampian Mountains were a major ice sheet divide (shown by the thick lines) with northwards moving ice (thin lines). For the Altar Stone to be transported from the Grampians, it would have to be captured within different glaciers, cross several major topographic barriers and ice sheet divides, and remain in ice for >750 km, which appears rather infeasible.

[REDACTED]

Scottish provenance for Wiltshire erratics.

It may be so that a Scottish provenance is unlikely for any given erratic. However, prehistoric Britain contained hundreds of thousands of erratics (far more than today). If you multiply a small probability with a large number of erratics, then an unlikely origin may become quite likely. It seems plausible to me that prehistoric people would choose the rarest erratics to build their monuments. So they may have 'cherry picked' their bluestones for rarity, unwittingly creating the puzzle that Stonehenge presents to scientists today.

The BRITICE Glacial Map (Clark et al. 2022) shows the occurrence of documented glacial erratics linked to their source (Figure 2). Erratic transport from the Southern Uplands, Grampian Mountains, and the Midland Valley to southern

[REDACTED]

Figure 1. Ice sheet reconstructions for Britain Ireland during the Last British Glaciation (Hughes

[REDACTED]

Figure 2. Glacial erratic sources and transport Scotland (Clark et al. 2022).

Britain has been documented in north Wales and northern England (Gibbard, 2007). However,

northwards moving ice transported Cairngorm and Grampian Mountain erratics **to** the Orcadian Basin rather than carrying material **away** from the basin.

Given these points, we view the glacial transport of the Altar Stone as highly unlikely, inconsistent with current observations, which leaves the alternative: the Altar Stone, a six-tonne-shaped sandstone megalith, to be anthropogenically transported >750 km from the Orcadian Basin to Salisbury Plain. This claim should be viewed in the context of other Stonehenge megaliths:

- The Sarsen stones (which weigh >25 tonnes) were transported from the West Woods, Marlborough, 25 km from Stonehenge (Nash et al. 2020).
- The Mynydd Preseli bluestones (which weigh 2 – 4 tonnes) were transported ~240 km to Salisbury Plain from SW Wales (Parker Pearson et al. 2021).

Neolithic Britons demonstrably had the technology and knowledge to carry multi-tonne cargo across challenging terrain. A transport distance of > 750 km for the six-tonne Altar Stone is unprecedented for Neolithic Europe. However, the 25 – 80 tonne granite blocks used in the contemporaneous (with Stage III of Stonehenge construction) Pyramid of Giza were shipped 900 km from Aswan to Giza (Lehner, 1997).

So, how did Neolithic people move the six-tonne Altar Stone? Let us consider an overland route first. While this is increasingly invoked for the Mynydd Preseli bluestones (Parker Pearson 2015; 2020) the transport of the Altar Stone from NE Scotland to Salisbury Plain would be orders of magnitude more challenging. Topographic barriers, such as the Grampian Mountains, Southern Uplands, Pennines, and North Yorkshire Moors, and the heavily forested landscape of Prehistoric Britain would have posed formidable obstacles (Godwin, 1975).

Moreover, rivers such as the Spey, Tay, Clyde, Tyne, Trent, and Severn would have to be variably navigated or crossed. Given these barriers and the fact that there is no direct evidence of how Neolithic Britons moved the megaliths to Stonehenge, we favour the marine transport of the Altar Stone from the Orcadian Basin as the most straightforward interpretation.

Evidence throughout Britain, Ireland, and Europe points towards a well-developed marine infrastructure that could have transported the Altar Stone. Neolithic people introduced the common vole (*Microtus arvalis*) to Orkney from Mainland Europe at ca. 3000 BC, implying the movement of cattle and goods on boats capable of carrying such cargo (Martínková, et al. 2013). Moreover, a Neolithic trade network of quarried stone artifacts between Britain, Ireland, and mainland Europe is recognised (Bradley et al. 2020). For example, jadeite axes found in Wessex are from the Italian Alps (Pétrequin et al. 2015). Therefore, there was a high degree of connectivity and trade in Prehistoric Britain, which, importantly, took place across open water.

Neolithic marine shipping was likely facilitated through sewn-plank boats capable of long-distance, open-water, sea-faring voyages (Van de Noort, 2014). The Hanson Log boat (dated 1500 BC) was found in Shardlow Derbyshire in a gravel pit along the River Trent in 1998. This 11 m long craft contained 500 kg cargo of Bromsgrove Sandstone, quarried from several km upstream (Crawshaw et al. 2013). This find dates to the later stages of Stonehenge's construction and demonstrates that Prehistoric Britons were shipping masonry.

Moreover, a Neolithic saddle quern, a large stone tool used for grinding flour, was discovered near Maiden Castle in Dorset and was determined to have a provenance in central Normandy

(Peacock and Cutler, 2010). This find is Britain's largest and heaviest Neolithic import (to date) and demonstrates open-water shipping. Archeological sites such as the Neolithic Howick House in Northumberland (Waddington and Bradley, 2003) and the Mesolithic Star Carr in Yorkshire (Mellars and Dark, 1998), show the ongoing inhabitation along southwards routes from the Orcadian Basin to Salisbury Plain. Moreover, the ca. 4000 BC Bouldnor Cliff settlement in The Solent preserves evidence of boat building and marine infrastructure (Strutt and Bates, 2004).

As you mentioned, why would Prehistoric Britons move an “ordinary” sandstone such an extraordinary distance? Many archaeologists consider NE Scotland and Orkney a nexus of trade, farming, fishing, and culture in Neolithic Britain (Bunting et al. 2022; Bayliss et al. 2017). Skara Brae on Orkney is one of Europe’s most complete Neolithic settlements and points towards a highly organised society with sophisticated infrastructure, building techniques, and economies (Sheridan and Davis, 1998; Ritchie, 2011). Furthermore, exotic materials such as jet, amber, Groove Ware pottery, and shale beads on Orkney suggest trade with mainland Scotland, eastern Britain, or even regions around the Baltic Sea (Renfrew and Bahn, 2016).

As we explore in the manuscript, isotopic evidence (Sr and Pb) from Salisbury Plain demonstrates the mobility of people (perhaps even from Scotland) and cattle throughout Neolithic Britain (Evans et al. 2022; Snoeck et al. 2018). The megaliths of Stonehenge were indeed “cherry-picked” in that they were collected from across Britain and represent a carefully curated selection of rocks. So, rather than an “ordinary” sandstone, it was sourced from the most accessible location to quarry such massive

	material near a population centre with good communication links.
--	---

“Statistical comparisons between crystalline basement terranes, ORS, and the Altar Stone, made using a Kolmogorov-Smirnov (KS) test, indicate that at 95% confidence, no distinction in provenance is evident between Altar Stone detrital zircon U–Pb ages and those from Laurentian basement; that is we accept the hypothesis that both samples are from the same age distribution (P-value >0.05) (Figure 3a).”

Here the authors misrepresent the concept of statistical hypothesis testing. Failure to reject a null hypothesis does NOT mean that said null hypothesis has been accepted. The outcome of a statistical hypothesis test (such as KS) depends on two things:

1. The degree to which the null hypothesis is false, and
2. Sample size. The second point is evident from Table 3 of the paper. For example, the D-value of “Dingle Peninsula” and “Anglo-Welsh Basin” is 0.10. This is exactly the same as the D-value for “Orcadian Basin” and “Altar Stone”. However, their p-values are different, at 0.33 and 0.00, respectively. The reason for this difference is that the “Altar Stone” sample is much smaller than the “Dingle Peninsula” and “Anglo-Welsh Basin” datasets.

As a second example, the D-value for “Laurentia” and “Altar Stone” is 0.22, which is higher than the D-value of 0.12 for “Ganderia” and “East Avalonia”. However, their p-value is lower. In other words, the D-values suggest that “Ganderia” and “East Avalonia” are more similar to each

Thank you for this point regarding statistical reporting.

Indeed, KS tests are sensitive to sample size. Therefore, as recommended by Vermeesch (2018), we implemented uncertainty calculations within multi-dimensional scaling space (Figure. 3a) to mitigate against sample-sized induced “Type-I errors”. Figure 3a shows a Bootstrap resampling (* 1000) and Procrustes rotation (Nordsvan et al. 2020) MDS plot of comparative zircon U–Pb datasets. On which your observations regarding D-values are apparent, similar “Dingle Peninsula” and “Anglo-Welsh Basin” datasets are proximal and orientated towards the same vector, as controlled by the age peaks. The uncertainty ellipses for “Laurentia” and “Ganderia” are small, given they are large datasets. Notably, the “Orcadian Basin” and “Altar Stone” ellipses overlap within 95% uncertainty and, along with “Laurentia” and “Svalbard”, are distinct from other datasets in that they occupy a discrete MDS area in Figure 3a.

In any case, to extend on this aspect, it is also possible to account for sample n in KS tests via Monte-Carlo resampling (Table 4 in Methods) (Gynn and Gehrels, 2010). By implementing Monte-Carlo resampling, the P-values (with two standard deviations) for the “Altar Stone” versus the “Orcadian Basin”, “Laurentia”, and Svalbard are **0.67 ± 0.16**, **0.11 ± 0.05** and **0.33 ± 0.10**, respectively. Thus, resampling yields consistent P-values in keeping with previous KS test results. Thus, our provenance interpretations for the Altar Stone and the Orcadian Basin remain robust with this further consideration of n .

“Statistical comparisons between **zircon ages from the Laurentian crystalline basement and the Altar Stone**, made using a Kolmogorov-Smirnov (KS) test, indicate that at a 95% confidence level, no distinction in provenance is

other than “Laurentia” and “Altar Stone”, but the p-values suggest the opposite!

evident between Altar Stone detrital zircon U–Pb ages and those from the Laurentian basement. **That is, we cannot reject the null hypothesis that both samples are from the same age distribution ($P>0.05$) (Figure 3a)."**

"The detrital zircon age spectra from Orcadian Basin ORS provide the closest match to the Altar Stone detrital ages ²⁴ (Figure 3; Extended Data Figure 8). A KS test on age spectra from the Altar Stone and the Orcadian Basin indicates that, at over 95% confidence, their age distributions are not different ($P> 0.05$) (Figure 3a)."

Text in the method reads:

"A two-sample Kolmogorov-Smirnov (KS) statistical test was implemented to compare compiled zircon age datasets. The test compares the maximum probability difference between two cumulative density age functions, evaluating the null hypothesis that two distributions are drawn from the same distribution based on a critical value dependent on the number of analyses in a distribution and the 95% confidence level.

"The number of zircon ages within the comparative datasets used here varies from the Altar Stone (56) to Laurentia (2469). Therefore, we also implemented Monte-Carlo resampling (1000 times) of the KS test, including the uncertainty on each age determination to calculate P-values and standard deviation (Table 4) based on the resampled synthetic distribution for each sample. Multi-dimensional scaling plots (MDS) for zircon datasets were created using the MATLAB script of Nordsvan et al. ²¹. Here, we adopted bootstrap resampling (>1000 times) with Procrustes rotation of KS values, **which outputs uncertainty ellipses at a 95% confidence level (Figure 3a). In MDS plots,**

	“stress” provides a goodness of fit indicator between dissimilarities in the datasets and distances on the MDS plot. Stress values below 0.15 are desirable ²¹. For the MDS plot in Figure 3a, the stress value is 0.043, indicative of an excellent fit.”
An extended discussion of this phenomenon is provided by Vermeesch (2018), who also points out that this problem undermines the validity of Satkoski et al. (2013)’s ad-hoc Likeness measure of dissimilarity (which is mentioned on line 488 of the manuscript and can be removed without consequence).	Thanks for this insight. Ok, we have removed the reference to Satkoski et al. 2013.
Line 464: “smallest” should be “largest” Line 497: “detectors” should be “mass analyser” Lines 429 and 530: “Lauren” should be “Laurin”	All edits made.

Reviewer Reports on the First Revision:

Referees' comments:

Referee #1 (Remarks to the Author):

I am happy that the points raised in my original review have been answered satisfactorily. On reading through the authors' responses to the points raised by the other two reviewers, it seems to me that they have done a good job in responding to these reviews as well. The addition of new apatite data from Orkney is very useful.

One minor (but important) editorial point: in line 141, the Dalradian is referred to as comprising Palaeozoic metasediments. Not so! It is mainly Neoproterozoic! Modify to read "Neoproterozoic to Lower Palaeozoic"?

Referee #2 (Remarks to the Author):

The authors have addressed my comments and answered my questions. I recommend the manuscript for publication.

Referee #3 (Remarks to the Author):

I had three main comments on the original version of this manuscript:

1. The paper's conclusions were mainly based on 56 zircon U-Pb dates, and little or nothing was done with the apatite and rutile U-Pb data, or the apatite Lu-Hf data. This is rather thin evidence for such a far-reaching conclusion.
2. The possibility that the Altar Stone is a former glacial erratic was not seriously considered.
3. Some mistakes were made in the statistical description of the data.

In this second round of review, I will focus on the extent to which these three points have been addressed.

1. In their response to my review, the authors have shown that the apatite U-Pb data for the Altar Stone are a good match to selected samples from the Orcadian Basin. However, this comparison only tells part of the story. What I had asked in my review was to complement the data with similar measurements from ORS samples from the Midland Valley and Wales. I suggested doing the same for the rutile and Lu-Hf data. This is not addressed in the rebuttal.

2. The authors repeat the reference to the BRITICE model, and conclude that “the glacial transport of the Altar Stone [is] highly unlikely”. I would say that a marine provenance is highly unlikely too. In statistical terms, the conclusions of this study should be based on the “odds ratio” of the two unlikely outcomes. Under this odds ratio, the unlikely glacial hypothesis may still remain plausible. BRITICE is a model. A sophisticated model perhaps, but still just a model, which could be wrong.

The authors argue that Neolithic Britons demonstrably had the technology to carry multi-tonne cargo over long distances. They write that “The Mynydd Preseli bluestones (which weigh 2 – 4 tonnes) were transported ~240 km to Salisbury Plain from SW Wales”

According to the BRITICE model, SW Wales was covered by glaciers. Would it not be possible that these bluestones were also transported in the direction of the Salisbury plain by ice, and that the remaining distance that had to be covered by the Neolithic Britons was 10s of kms instead of 100s of kms?

Finally, the authors claim that the ORS of the Orcadian Basin is not an “ordinary” sandstone, but “was sourced from the most accessible location to quarry such massive material near a population centre with good communication links.” This raises the question why there is only one megalith from the Orcadian Basin.

In conclusion, marine transport of the Altar Stone raises more questions than answers. I don’t mind if the authors express their preference for a marine transport mechanism. However, I haven’t seen any evidence that rules out a glacial origin completely. A glacial origin may be unlikely, but unlikely things happen all the time.

3. The revised version of the manuscript addresses some of my concerns but not all of them. Highlighting some of the changes:

“That is, we cannot reject the null hypothesis that both samples are from the same age distribution ($P > 0.05$) (Figure 3a).”

-> This is correct

“A KS test on age spectra from the Altar Stone and the Orcadian Basin indicates that, at over 95% confidence, their age distributions are not different ($P > 0.05$) (Figure 3a).”

-> This is wrong. I suggest rephrasing as follows:

“A KS test on age spectra from the Altar Stone and the Orcadian Basin fails to reject the null hypothesis that they are derived from the same underlying distribution ($P > 0.05$) (Figure 3a).”

“Therefore, we also implemented Monte-Carlo resampling (1000 times) of the KS test, including the uncertainty on each age determination to calculate P-values and standard deviation”

-> Confidence intervals have a precise meaning, which does not fit the Monte-Carlo resampling method described here. Also, please do not calculate “standard deviations of p-values” as it will make any statistician cringe.

I would suggest removing this bit, as well as the p-values in Table 3 and the entire Table 4. As the authors point out in the manuscript, the MDS configuration successfully avoids all the pitfalls surrounding hypothesis tests. The bootstrapped sensitivity regions are a useful addition as well.

Author Rebuttals to First Revision:

Reviewer 1	Reply
I am happy that the points raised in my original review have been answered satisfactorily. On reading through the authors' responses to the points raised by the other two reviewers, it seems to me that they have done a good job in responding to these reviews as well. The addition of new apatite data from Orkney is very useful. One minor (but important) editorial point: in line 141, the Dalradian is referred to as comprising Palaeozoic metasediments. Not so! It is mainly Neoproterozoic! Modify to read "Neoproterozoic to Lower Palaeozoic"?	We thank Reviewer 1 for their insightful points once again. We have amended the description of the Dalradian Supergroup so it now reads: “Situated between the Great Glen Fault to the north and the Highland Boundary Fault to the south, the terrane comprises Neoproterozoic to Lower Palaeozoic metasediments termed the Dalradian Supergroup ²⁷, which are intruded by a compositionally diverse suite of early Palaeozoic granitoids and gabbros (Figure 4a).”

Reviewer 2	Reply
The authors have addressed my comments and answered my questions. I recommend the manuscript for publication.	Thank you for your helpful comments and for recommending our work for publication

Reviewer 3	Reply
I had three main comments on the original version of this manuscript:  1. The paper's conclusions were mainly based on 56 zircon U-Pb dates, and little or nothing was done with the apatite and rutile U-Pb data, or the apatite Lu-Hf data. This is rather thin evidence for such a far-reaching conclusion. 2. The possibility that the Altar Stone is a former glacial erratic was not seriously considered. 3. Some mistakes were made in the statistical description of the data 	Thank you for your additional review of our work. We addressed those points listed here in the previous round of revisions.  1. The work is based on over 250 U-Pb analyses of zircon, rutile, and apatite from the Altar Stone and >130 apatite U-Pb analyses from the Orcadian Basin. 2. This is incorrect; glacial transport was considered and is inconsistent with ice flow directions (based on the available geological evidence). 3. This was addressed in the previous review round, including additional tests that reinforced our interpretations.
 1. In their response to my review, the authors have shown that the apatite U-Pb data for the Altar Stone are a good match to selected samples from the Orcadian Basin. However, this comparison only tells part of the story. What I had asked in my review was to complement the data with similar measurements from ORS samples from the Midland Valley and Wales. I suggested doing the same for the rutile and Lu-Hf data. This is not addressed in the rebuttal. 	We provided additional data from apatite that strengthened our interpretation, which is already strong given the current zircon, apatite, and rutile dataset. The match between the Altar Stone and the Orcadian basin is robust for any statistical measure.
 2. The authors repeat the reference to the BRITICE model, and conclude that "the glacial transport of the Altar Stone [is] highly unlikely". I would say that a marine provenance is highly unlikely too. In statistical terms, the conclusions of this study should be based on the "odds ratio" of the two unlikely outcomes. Under this odds ratio, the unlikely glacial hypothesis may still remain plausible. BRITICE is a model. A sophisticated model perhaps, but still just a model, which could be wrong. 	We are unsure what a marine provenance means in the context of the referee's statement. If it refers to the original depositional setting, then the literature on the Orcadian Basin demonstrates that it was deposited in continental fluvial, lacustrine, and aeolian environments (Kendall, 2017). If, as we presume, the reviewer is referring to marine transportation of the stone block, all we need to state here is that the current geological evidence and ice flow model are inconsistent with a glacial transport mechanism for the Altar Stone and, thus, a probabilistic consideration of the data indicates marine transport is more likely.

The authors argue that Neolithic Britons demonstrably had the technology to carry multi-tonne cargo over long distances. They write that “The Mynydd Preseli bluestones (which weigh 2 – 4 tonnes) were transported ~240 km to Salisbury Plain from SW Wales”	“The Mynydd Preseli bluestones (which weigh 2 – 4 tonnes) were transported ~240 km to Salisbury Plain from SW Wales” – this is based on published literature, which documents Neolithic quarrying sites for the Mynydd Preseli bluestones (Parker-Pearson et al. 2015). The reviewer appears to confuse igneous “Bluestones”, sourced from Wales, with the Altar Stone, which has a distinct provenance. A multi-glacier pathway is feasible, but this neglects much of the known archaeology. Importantly, there is no evidence of ice transport on the Altar Stone.
According to the BRITICE model, SW Wales was covered by glaciers. Would it not be possible that these bluestones were also transported in the direction of the Salisbury plain by ice, and that the remaining distance that had to be covered by the Neolithic Britons was 10s of kms instead of 100s of kms?	
Finally, the authors claim that the ORS of the Orcadian Basin is not an “ordinary” sandstone, but “was sourced from the most accessible location to quarry such massive material near a population centre with good communication links.” This raises the question why there is only one megalith from the Orcadian Basin	It would be speculation to comment further on this aspect.
In conclusion, marine transport of the Altar Stone raises more questions than answers. I don’t mind if the authors express their preference for a marine transport mechanism. However, I haven’t seen any evidence that rules out a glacial origin completely. A glacial origin may be unlikely, but unlikely things happen all the time.	We present our preferred interpretation based on a robust geochronology study. We agree that the findings and interpretations open up new areas of study, which we regard as both exciting and important. Philosophically, what is unseeable is unknowable; hence, we are left to interpret the available evidence.
3. The revised version of the manuscript addresses some of my concerns but not all of them. Highlighting some of the changes: “That is, we cannot reject the null hypothesis that both samples are from the same age distribution ($P > 0.05$) (Figure 3a).” -> This is correct	Thank you for your advice on this subtle rewording. We have rephrased the manuscript following your recommendation. The text now reads:
“A KS test on age spectra from the Altar Stone and the Orcadian Basin indicates that, at over 95% confidence, their age distributions are not different ($P > 0.05$) (Figure 3a).”	“A KS test on age spectra from the Altar Stone and the Orcadian Basin fails to reject the null hypothesis that they are derived from the same underlying distribution (KS test: $P > 0.05$) (Figure 3a).”

-> This is wrong. I suggest rephrasing as follows: “A KS test on age spectra from the Altar Stone and the Orcadian Basin fails to reject the null hypothesis that they are derived from the same underlying distribution ($P > 0.05$) (Figure 3a).”	
“Therefore, we also implemented Monte-Carlo resampling (1000 times) of the KS test, including the uncertainty on each age determination to calculate P-values and standard deviation” -> Confidence intervals have a precise meaning, which does not fit the Monte-Carlo resampling method described here. Also, please do not calculate “standard deviations of p-values” as it will make any statistician cringe.	We provided the Monte-Carlo resampling (Gehrels 2012, 2014) specifically for this reviewer to demonstrate further that sample n was not leading to a change in statistical interpretation. Moreover, this aspect was always covered in the MDS plot with bootstrapped uncertainty. In short, the final interpretation result is not dependent on sample n.
I would suggest removing this bit, as well as the p-values in Table 3 and the entire Table 4. As the authors point out in the manuscript, the MDS configuration successfully avoids all the pitfalls surrounding hypothesis tests. The bootstrapped sensitivity regions are a useful addition as well.	KS tests are customarily shown on a contingency table, and we choose to retain these. The KS test results are consistent with the MDS plot, and we show both in the methods section for the interested reader.